# Navigating the Maps: Euclidean vs. Road Network Distances in Spatial Queries

Pornrawee Tatit [1], Kiki Adhinugraha [2] and David Taniar [1,*]

1 Faculty of Information Technology, Monash University, Melbourne, VIC 3800, Australia; yow.tatit@monash.edu
2 Department of Computer Science and Information Technology, La Trobe University, Melbourne, VIC 3086, Australia; k.adhinugraha@latrobe.edu.au
* Correspondence: david.taniar@monash.edu

**Abstract:** Using spatial data in mobile applications has grown significantly, thereby empowering users to explore locations, navigate unfamiliar areas, find transportation routes, employ geomarketing strategies, and model environmental factors. Spatial databases are pivotal in efficiently storing, retrieving, and manipulating spatial data to fulfill users' needs. Two fundamental spatial query types, k-nearest neighbors (kNN) and range search, enable users to access specific points of interest (POIs) based on their location, which are measured by actual road distance. However, retrieving the nearest POIs using actual road distance can be computationally intensive due to the need to find the shortest distance. Using straight-line measurements could expedite the process but might compromise accuracy. Consequently, this study aims to evaluate the accuracy of the Euclidean distance method in POIs retrieval by comparing it with the road network distance method. The primary focus is determining whether the trade-off between computational time and accuracy is justified, thus employing the Open Source Routing Machine (OSRM) for distance extraction. The assessment encompasses diverse scenarios and analyses factors influencing the accuracy of the Euclidean distance method. The methodology employs a quantitative approach, thereby categorizing query points based on density and analyzing them using kNN and range query methods. Accuracy in the Euclidean distance method is evaluated against the road network distance method. The results demonstrate peak accuracy for kNN queries at $k = 1$, thus exceeding 85% across classes but declining as k increases. Range queries show varied accuracy based on POI density, with higher-density classes exhibiting earlier accuracy increases. Notably, datasets with fewer POIs exhibit unexpectedly higher accuracy, thereby providing valuable insights into spatial query processing.

**Keywords:** spatial query processing; open street map (OSM); map navigation; GIS; spatial data; location processing; point of interest (POI)





## 1. Introduction

The utilization of spatial data in mobile applications has experienced significant growth, thereby enabling users to search for locations easily, navigate unfamiliar areas, identify transportation routes, employ geomarketing techniques, and model environmental factors [1]. To illustrate, users can search for nearby hospitals using map applications or GPS navigation (Figure 1). To meet users' requirements, spatial databases play a vital role in storing, retrieving, and manipulating spatial data. These databases are designed to operate efficiently and effectively with spatial data, thereby ensuring that users can quickly and accurately retrieve the data. There are two main types of spatial queries: k-nearest neighbors (kNN) and range search. K-nearest neighbors aim to find a specific number of nearest points of interest (POIs) based on the user's location, while range search retrieves objects within a certain distance radius. For instance, if a user searches for the six nearest hospitals, the user's location serves as a query point, and the map application will provide

only six hospitals closest to the query point. On the other hand, range search differs from kNN, as it is limited by distance rather than the number of spatial objects. For example, a user can search for hospitals within a 10 km radius, and the application will return all the hospitals within that range.

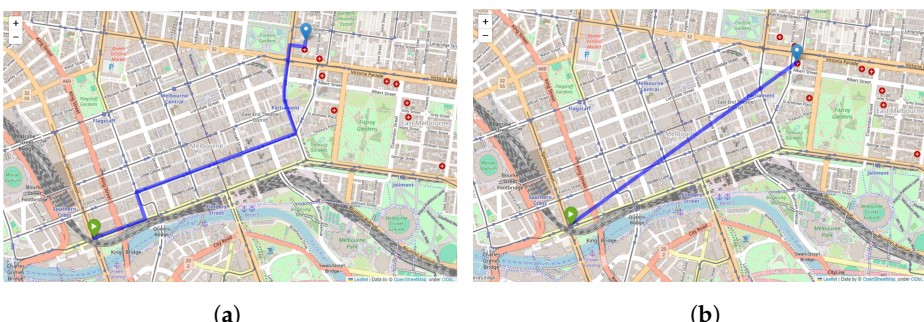

(**a**)          (**b**)

**Figure 1.** Search for the nearest hospital. (**a**) Result from road network distance. (**b**) Result from Euclidean distance.

As the use of map applications continues to increase, spatial databases encounter the challenge of effectively managing a substantial amount of spatial queries. This poses a considerable computational burden in retrieving spatial objects due to the need for the database to measure the distance between the query point and all spatial objects, thus identifying those that align with the user's requirements. This process results in heightened computational costs.

The road network distance method is employed to ensure the accuracy of retrieved spatial objects. This method utilizes Dijkstra's algorithm to find the shortest path based on the actual road distance. The database can accurately retrieve the desired objects by calculating the shortest route from the query point to the POIs. However, using Dijkstra's algorithm introduces high computational time due to the complexity of finding the shortest distance. The time complexity of Dijkstra's algorithm for cost functions based on vertices is approximately $O(|E| + |V|log|V|)$ [2], where $E$ represents the set of edges in the graph, and $V$ represents the set of vertices. Moreover, when dealing with an extensive road network, the calculations required for determining the shortest paths can impose a considerable demand on both time and memory resources [3]. In scenarios such as kNN, where the total number of objects far exceeds k, computing the shortest paths for all objects to identify the k-nearest neighbors becomes impractical [4]. This presents challenges, as it conflicts with the primary objective of the database, which is to enhance the retrieval process by ensuring efficiency and accuracy in data retrieval. The Euclidean distance method is a better choice to address this issue and reduce the computational complexity of retrieving POIs from spatial databases. This method calculates the straight-line distance between the query point and spatial objects without finding the shortest path. As a result, the computational complexity of calculating Euclidean distance is relatively simple, with a constant time complexity of $O(1)$ [2].

Additionally, Open Source Routing Machine (OSRM) has emerged as a high-performance routing service designed for use with geographic data. As an open-source project, OSRM provides routing functionality, thereby allowing users to extract and calculate the shortest or fastest route between locations using Dijkstra's algorithm. The capabilities of OSRM further enhance the efficiency and accuracy of spatial query processing in spatial databases.

Therefore, the primary objective of this study is to assess the accuracy of the Euclidean distance method for retrieving points of interest (POIs) by comparing it with the road network distance method, which utilizes the OSRM to extract distances. The evaluation will be conducted in various scenarios to observe the accuracy of the Euclidean distance method and to analyze factors that may influence its accuracy rate. This analysis provides valuable insights regarding the reliability and effectiveness of the Euclidean distance method in spatial query processing.

## 2. Related Work

### 2.1. Spatial Queries in Euclidean Distance

Spatial data objects represent a type of data that encompasses multiple dimensions. Over the past few years, there has been a growing utilization of this data type in various applications, including geographic information systems (GIS), computer-aided design, and computer vision [5]. The commonly employed methods for searching for spatial objects include k-nearest neighbors (kNN) and range query techniques.

2.1.1. K-Nearest Neighbor Queries in Euclidean Distance

The effective execution of nearest neighbor (NN) queries, which holds special significance in the field of geographic information systems (GIS) [6], is essential when considering the commonly used query type known as k-nearest neighbors (kNN). The kNN query involves finding the potential neighbors closest to a given query point [6,7]. In retrieving points of interest (POIs) using k-nearest neighbors, the distance between the query point and the POIs is calculated by measuring the straight-line distance from the query point's coordinates to the coordinates of each POI. The resulting POIs are then ordered based on their distance from the query point, thereby enabling the selection of only the top k-ranked POIs from the results. To illustrate, Figure 2 depicts a map diagram with weighted lines representing the actual distance of the road. The distances are measured using the Euclidean distance method, as shown in Figure 3. As a result, the retrieved POIs from the Euclidean distance method would be P6, P2, and P7, as indicated in Figure 4.

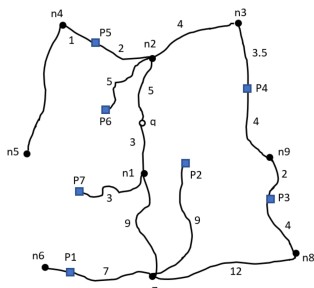

**Figure 2.** Road and locations with lines represent roads with nodes and the distance and locations.

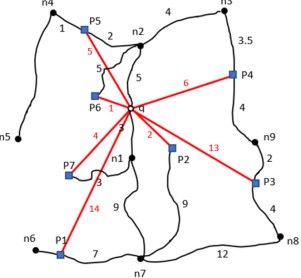

**Figure 3.** Red lines show the measurement of the distance using the Euclidean distance method.

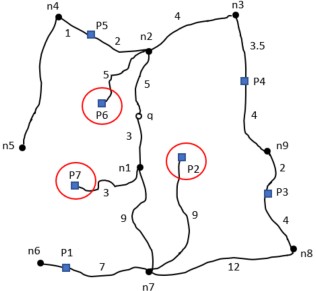

**Figure 4.** The result of kNN query using the Euclidean distance method.

Fukunaga and Narendra [8] proposed an effective method for computing k-nearest neighbors. Their approach involves a hierarchical decomposition of design samples into disjoint subsets. To achieve this, they applied the branch and bound algorithm, which is a well-regarded tree search technique [9–11]. This algorithm is known for efficiently searching through the resultant groups.

Expanding the scope of k-nearest neighbors (kNN) query processing, Taniar and Rahayu [7] discussed the extension of k-nearest neighbors (kNN) query processing to include obstacles in Euclidean space, known as obstacle nearest neighbors (ONNs). ONNs involve constructing a visibility graph, representing possible paths around obstacles, and using a disk-based approach with a restriction method to process the kNN queries while considering obstacle distances efficiently. The method iteratively refines search spaces based on obstacle distances to optimize query processing.

Additionally, various types of nearest neighbor queries have been studied, including aggregate nearest neighbor (ANN) [12], reverse nearest neighbor (RNN) [13], and group nearest neighbor (GNN) [14]. However, our study focuses explicitly on the k-nearest neighbor (kNN) queries.

### 2.1.2. Range Queries in Euclidean Distance

Range queries are an essential query method in database systems [15]. Building upon the concept of a point query, the area around the query point is expanded in range queries, thus forming an area of query scope. Range queries involve three primary components: points of interest, the query point, and the query scope. To retrieve the relevant objects, range queries utilize intersection and containment operations. The distance between the query point and each point of interest is calculated by measuring the straight-line distance based on their respective coordinates. The resulting distances are then compared with the range specified by the query scope. Objects of interest that fall within this range become the query result [16]. For example, considering an area of query scope as depicted in Figure 5, the points of interest (POIs) located within that area, such as P6, P2, and P7, would be retrieved as the results of the range query (Figure 6).

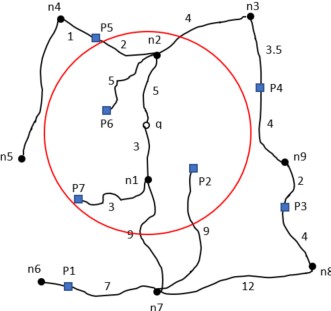

**Figure 5.** Red circle represent the area scope for range queries.

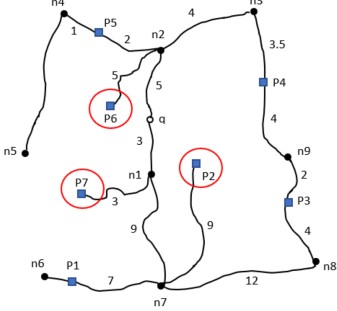

**Figure 6.** The result of range query using the Euclidean distance method.

In the realm of range queries, a research study explored continuous range search algorithms, thereby highlighting the limitations of traditional methods designed for static points. The paper emphasizes the inadequacy of these approaches for mobile users in motion. In response, the authors propose two methods for continuous range search, with the first focusing on Euclidean distance [17].

Another research study by Pfoser, Jensen, and Theodoridis [18] evaluated three access methods: the R-Tree, spatiotemporal R-tree, and trajectory bundle tree. The performance study involved experiments with spatial range, as well as navigational and combined queries.

*2.2. Spatial Queries in Road Network Distance*

According to [7], a spatial road network is fundamentally an interconnected system of roads, which is illustrated as a network consisting of edges (links) and vertices (nodes). An edge or link serves as a connection between two vertices or nodes. Each link connecting two adjacent nodes forms a segment (an unbroken link from one node to another) without passing through any additional nodes. The distance associated with each segment is specified and referred to as the weight of that segment. The spatial road network can be utilized for conducting k-nearest neighbor (kNN) and range queries using two different approaches: restriction and expansion [7,19].

2.2.1. K-Nearest Neighbor Queries in Road Network Distance

The algorithm for kNN on spatial road networks employs two approaches, as aforementioned. The first approach, focused on restriction kNN, is the incremental Euclidean restriction (IER) algorithm. The IER approach involves setting an upper limit and reducing the search area. The second approach, centered on expansion kNN, is called incremental network expansion (INE). The algorithm initiates network expansion from the query point, thus examining entities in the order they are encountered [19].

Abeywickrama, Cheema, and Taniar [4] experimented with evaluating kNN query algorithms on road networks, including incremental network expansion (INE) [19], incremental Euclidean restriction (IER) [19], distance browsing [20], route overlay and association directory (ROAD) [21,22], and G-tree [23,24].

In their research paper, Shahabi, Kolahdouzan, and Sharifzadeh [25] proposed a new approach for handling kNN queries in road networks, thereby catering to stationary and mobile query points. Their approach, named Road Network Embedding (RNE), involves converting a road network into a higher-dimensional space, thus allowing the use of more straightforward distance functions.

Kolahdouzan and Shahabi [26] introduced a novel approach to address the criteria for spatial query in spatial network databases (SNDBs) by transforming the challenge of distance computation in a vast network into the problem of computing distances in numerous smaller networks, which are supplemented by additional table lookups. The central concept behind their approach, Voronoi-based Network Nearest Neighbor (VN3), involves initially partitioning an extensive network into smaller, more manageable regions.

In the research conducted by Mouratidis, Yiu, Papadias, and Mamoulis [27], the focus was on continuous k-nearest neighbor (CkNN) queries in transportation networks, where data objects and queries move within a road network. While existing methods have addressed snapshot queries, continuous monitoring of CkNN queries in road networks has not yet been explored. The study introduces incremental monitoring and group monitoring algorithms to compute and update CkNN query results efficiently in real time, thereby considering fluctuations in edge weights due to factors like changing traffic conditions. The proposed algorithms aim to handle dynamic and unpredictable scenarios, such as finding the k closest taxi customers for a free taxi in terms of traveling time.

2.2.2. Range Queries in Road Network Distance

In the paper by [16], it is asserted that range queries, rooted in defining a region according to a specified radius or distance, can be more broadly and flexibly referred to as "region queries". These queries involve three essential components: the object of interest, the query point (representing the user's query location), and the designated region. Taniar and Rahyau [16] categorized region queries into six types: traditional region queries, approximate region queries, constrained region queries, outer/inner fence object queries, and inverse range queries.

Traditional regions establish boundaries to extract candidate objects. Approximate region queries address situations where users may inaccurately specify the query radius due to unfamiliarity with the area. A constrained region query incorporates additional constraints, such as spatial, temporal, kNN, or others, along with the query radius. Clustered objects region queries consider objects with spatial relationships, thereby aiming to identify clusters or groups of objects within a specified region. Inverse range queries involve multiple query points or objects and seek candidate objects that contain all query points within a specified range.

Regarding range queries in road networks, two approaches are identified: restriction and extension [19]. The restriction approach for range queries is called range Euclidean restriction (RER). This method conducts a range query to find objects within a specified Euclidean distance from the query point. While maintaining the Euclidean lower bound property, it reduces false hits by performing a single network expansion. The algorithm efficiently refines results using sorted lists, segment checks, and comparisons, thereby considering boundary segments that may exceed the query threshold. On the other hand, the expansion approach for range queries is called range network expansion (RNE). This algorithm identifies qualifying segments within a specified network range from the query point and retrieves associated data entities. It optimizes by dividing qualifying segments (QSs) into sets corresponding to R-tree entries, thereby minimizing comparisons as it descends the tree. The algorithm ensures I/O optimality and handles scenarios where the QSs exceed memory capacity through efficient methodologies and optimizations.

A study in 2.1.2 [17] explored the continuous range search within mobile navigation, thus explicitly focusing on user mobility. The study proposed a second approach emphasizing network distance over Euclidean distance, thereby aiming for enhanced efficiency in practical applications. The outlined method involves selecting an arbitrary path from a road map, segmenting it, and utilizing range network expansion (RNE) to identify entities within a specified range while considering Euclidean distance.

*2.3. Existing Research*

One research study explored the relationship between transport costs and road distance. The researchers employed a correlation factor to estimate the actual road distance by considering the Euclidean distance. They evaluated the accuracy of this approximation by calculating the average difference between the estimated and actual distances [28]. The conclusion highlights the development of a linear regression model for Euclidean distances, thereby demonstrating its superior reliability compared to existing literature models. Two adjustment methods are discussed: one based on mean ratios and another utilizing regression equations. While more computationally intensive, the latter provides closer-to-reality results, but caution is urged regarding its constant, particularly for smaller distances. Notably, the study did not provide explicit information regarding the precision of the Euclidean distance. Instead, it introduced a regression model to approximate the road distance and compared it with the actual distance.

In another research study, the accuracy of the Euclidean distance was compared with the road network distance. The study investigated the performance of two query processing methods: the benchmark method, which utilized the road network distance, and the Euclidean decider, which employs the Euclidean distance [29]. The paper suggests using Euclidean distance as a fast and computationally less intensive alternative for estimating

road network distance. The authors argue that, despite potential differences, Euclidean distance-based solutions can provide acceptable accuracy for specific queries, such as group nearest neighbor (GNN) queries. Experiments involved comparing two GNN query processing methods using road network distance as a benchmark and Euclidean decider (using Euclidean distance). Actual road network graphs from cities were utilized, thereby creating scenarios with scattered or clustered query points and points of interest (POIs). Parameters such as the number of query points, POI density, the value of k, and road network were varied to assess performance. Euclidean decider's accuracy was evaluated by comparing distances with the benchmark results. The experiments aimed to determine the effectiveness of Euclidean distance in achieving a balance between accuracy and computational efficiency for GNN queries in transportation services. The study's results [29] indicate that the Euclidean metric achieved an accuracy level exceeding 74.7% for kNN queries and exhibited even higher accuracy for general group nearest neighbor (GNN) queries, which ranged from 86% to 95% when considering two query points.

In a related work, Kim, A. Hossain, A.-A. Hossain, and Chang proposed the Hilbert-order based star expansion cloaking algorithm (H-star) [30]. This algorithm is designed to optimize query processing cost by considering cloaking regions for a group of queries, thereby reducing the number of boundary points and, consequently, the query processing cost. The proposed H-Star algorithm has been extended to include query processing algorithms for k-nearest neighbor (kNN) and range queries.

Another research study proposed a method for road network extraction from high-resolution synthetic aperture radar (SAR) images. The approach involved constructing road networks using smooth, crosslinked curves with determined functions, thereby enabling mathematical descriptions for road segments. The method employed a multiplicative Duda operation for line feature responses, and nonroad detection techniques were introduced to reduce false positives. Binary image decomposition and polynomial curve fitting were then used to linearize road segments, and network optimization was achieved through geometric constraints [31].

In a study by Arbelaez, Mehta, O'Sullivan, Quesada, and Sasmaz [32], a valuable methodology was provided for classifying exchange sites and customers into three geotypes (rural, suburban, and urban) based on precise customer locations rather than relying solely on the number of connected customers. The classification was defined by household density per square kilometer, with rural areas having up to 10 households/km², suburban areas ranging from 11 to 500 households/km², and urban areas exceeding 501 households/km². A distribution analysis in Ireland highlighted a predominance of rural and suburban regions, thereby offering valuable insights for refining the classification of query points or locations in diverse scenarios.

Lastly, Boyacı, Dang, and Letchford [3] studied vehicle routing problems (VRPs) on road networks, called Steiner VRPs. Unlike traditional VRPs, where customers and depots are nodes in a complete graph, Steiner VRPs consider road networks, thereby introducing challenges in terms of distance computation. The authors proposed a three-phase heuristic approach: (1) create an approximation using Euclidean distances multiplied by a constant; (2) solve the approximated instance; and (3) convert the solution to the original instance. Computational experiments were conducted on Steiner versions of the travelling salesman problem (TSP) and capacitated VRP (CVRP) using global road network data from twelve cities. The result of this research showed that the use of Euclidean distances instead of acctual road distances was found to yield acceptable results for the Steiner travelling salesman problem (TSP) and Steiner capacitated vehicle routing problem (CVRP), particularly when only a tiny proportion of nodes require service.

## 3. Methodology

This research employed a quantitative approach to evaluate the accuracy of the Euclidean distance method. Initially, a dataset was compiled, including hospital and ambulance addresses as points of interest (POIs) and residential addresses as query points. After

data collection, the dataset was divided into hospital and ambulance addresses. Subsequently, a data exploration process was undertaken, thereby revealing that only five cities had sufficient points of interest to experiment. Therefore, 30 query points were randomly selected from these five cities for the research. This careful process ensured a thorough evaluation of the accuracy of the Euclidean distance method in the selected urban areas is studied. The query points were further categorized based on quantitative criteria, primarily the density of points of interest (POIs), which was determined using attribute weights for precise calculations. This classification facilitated a systematic analysis of the data.

In the subsequent step, both Euclidean distance and the road network distance method were employed for POI retrieval, thereby utilizing kNN and range search techniques. The accuracy of the Euclidean distance method was then assessed by comparing its outcomes with those derived from the road network distance method, which served as a reliable quantitative benchmark. The road network distance method provided accurate distance calculations based on actual road networks.

This study followed a systematic approach as depicted in Figure 7, where the methodology flow chart illustrates the step-by-step process. By conducting thorough quantitative analyses and carefully comparing the results, this research aimed to evaluate the accuracy of the Euclidean distance method in retrieving spatial data, thereby ensuring a comprehensive and reliable assessment.

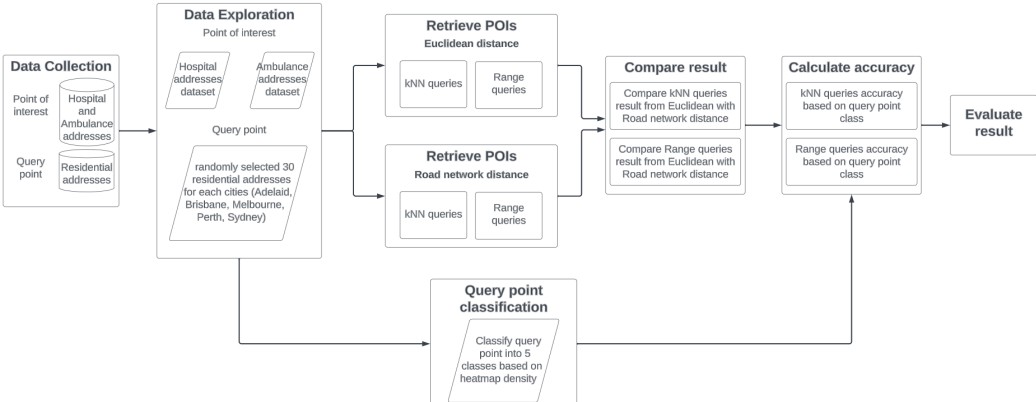

**Figure 7.** Methodology process with arrows representing the sequential flow of the process and the process explanation shown in each box.

### 3.1. Data Source

The dataset for hospital and ambulance addresses was obtained from OpenStreetMap (OSM), which is a comprehensive database containing information on hospitals and ambulances across various Australian states. In contrast, a segment of the residential addresses data originated from the Geoscape Geocoded National Address File (G-NAF) [33], which adheres to the Australian Statistical Geography Standard (ASGS) [34] and encompasses addresses located within the residential mesh blocks (MBs) framework. Specifically, residential addresses were randomized, and ten addresses from each Statistical Area Level 2 (SA2s) area were selected, thereby ensuring a representative sample for the study.

To classify the query points, the mesh blocks shapefile was employed to assess their characteristics based on the density of the POIs. Mesh blocks, defined by the Australian Bureau of Statistics [34], are small geographical areas designed for statistical and data aggregation purposes. They offer a structured way to organize and analyze fine-grained data. These mesh blocks served as a crucial framework for categorizing the query points, thereby enabling the evaluation of POI density near these points.

In our research, the residential address dataset was deliberately structured by mesh blocks, thereby ensuring the absence of duplicate addresses. This strategic alignment is a precautionary measure to eliminate the potential bias that might arise from duplicate addresses within the dataset. By selecting the query point in this manner, the experimental

data were able to accurately evaluate the accuracy of the Euclidean distance method in identifying the closest hospital to a given residential address.

*3.2. Data Exploration*

The POIs dataset was divided into two categories: hospitals and ambulances. The hospital dataset exclusively comprised hospital-related information, while the ambulance dataset contained data about ambulances. As a result, separate queries were performed for each dataset type.

Due to the significant volume of POIs resulting from the dataset split, only five cities were suitable for conducting the queries. As outlined in Table 1, specific areas exhibited a limited number of POIs, whether hospitals or ambulances, thereby potentially affecting the reliability of results. To ensure the robustness of our findings, we utilized a heatmap graph (Figure 8) illustrating the density of POIs. This visualization provided valuable insights into the spatial distribution of POIs, thereby guiding the selection of cities with a sufficient concentration. The heatmap analysis, shown in Figure 8, represents the density of point distribution through colored map areas, with a gradient from brighter to unshaded indicating higher point concentrations. This informed the inclusion of cities where the density of hospitals and ambulances met the experiment's criteria. Consequently, the experiment focused on cities with numerous POIs: Adelaide, Brisbane, Melbourne, Perth, and Sydney.

**Table 1.** Number of hospitals and ambulances for each state.

| State | Hospital | Ambulance |
|---|---|---|
| Northern Territory | 6 | 2 |
| Australian Capital Territory | 2 | 8 |
| Tasmania | 3 | 34 |
| New South Wales | 168 | 237 |
| Victoria | 38 | 219 |
| South Australia | 14 | 111 |
| West Australia | 24 | 155 |
| Queensland | 26 | 250 |

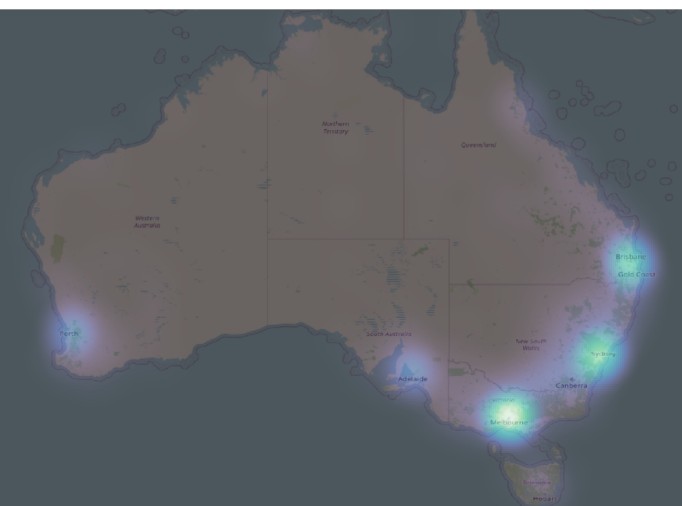

**Figure 8.** POIs density heatmap.

Limiting our study to just five cities with a higher density of hospitals and ambulances aimed to provide a more comprehensive and accurate assessment of the correctness of the Euclidean distance method in identifying the nearest hospitals or ambulances to a given residential address. This approach ensured that the experimental results were comprehensive and accurate. Furthermore, including only five cities with a higher density

of hospitals and ambulances introduced the necessary variance in the results, thereby allowing for a comparison of the accuracy between the Euclidean and road network distance methods. This methodological choice enhances the thoroughness and accuracy of our assessment of the Euclidean distance method's correctness in identifying the nearest hospitals or ambulances for residential addresses, thereby ensuring our experimental findings' overall comprehensiveness and accuracy.

### 3.3. Classification

This section explains in detail the methodology for classifying query points based on POI density.

### 3.3.1. Classification Criteria

The classification of query points involves four criteria: buffer creation, overlapping with mesh block, weight calculation, and using the mode function in QGIS for query point categorization based on the respective weights.

#### Buffer Creation

The first criterion is buffer creation, where spatial areas are generated around each point of interest to classify the query points. Buffers are generated based on specified distances. This study employed four different buffer distances: 1 km, 5 km, 10 km, and 15 km. These buffer distances were selected according to previous research on healthcare accessibility and surveys [35–37]. These studies provide insights into acceptable distances for accessing healthcare from households and examine the relationship between distance and utilization of health services.

#### Overlapping with Mesh Block

After creating the POI buffer with the specified distances, the criterion of overlapping with mesh block comes into play. Mesh blocks containing the query points are retrieved. POIs buffers are overlaid with mesh block shapes (Figure 9 with the mesh block in red shape inside the red square and POIs buffer in green and purple). This overlapping process determines the number of weight attributes used in calculating the density of POIs. The calculation involves counting the POIs buffer distances intersecting each mesh block, thereby contributing to the overall density calculation. The resulting count is stored in the database column (Figure 10).

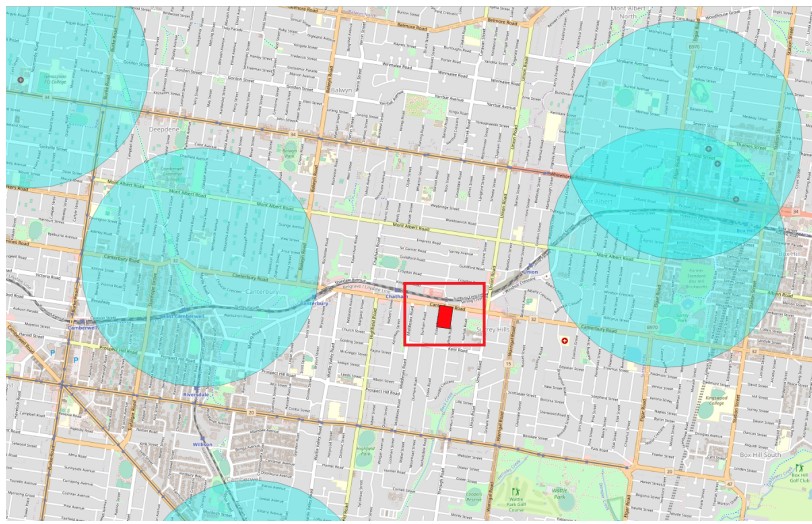

(a)

**Figure 9.** *Cont*.

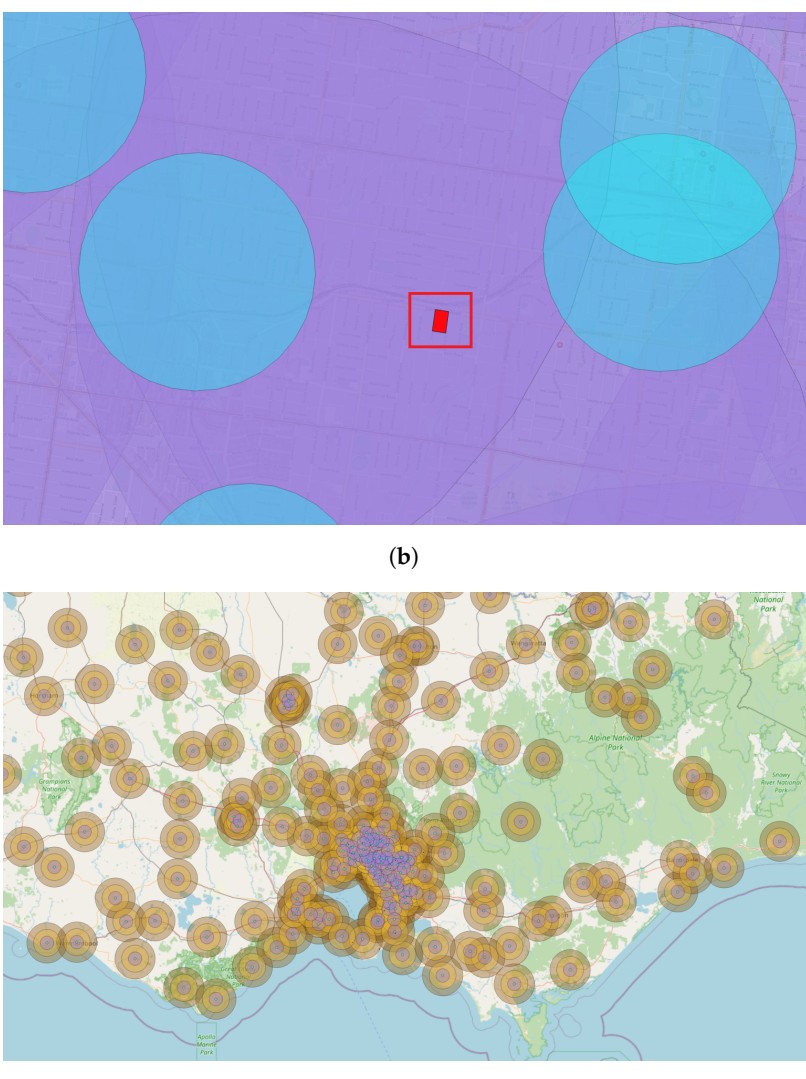

**Figure 9.** Mesh blocks with overlaid POI buffer. (**a**) Mesh block with 1 km POI buffer; (**b**) Mesh block with 1 km and 5 km POI buffers; (**c**) POI buffers (1 km, 5 km, 10 km, and 15 km).

| | mb_code21 | geom | buffer_1 km | buffer_5 km | buffer_10 km | buffer_15 km |
|---|---|---|---|---|---|---|
| 1 | 20023570000 | MULTIPOLYGON (((145.03832298865717 -37.76737625227523, 145.03835 | 0 | 2 | 16 | 28 |
| 2 | 20047560000 | MULTIPOLYGON (((145.01658255418653 -37.934599391819596, 145.0163 | 0 | 3 | 6 | 14 |
| 3 | 20064790000 | MULTIPOLYGON (((145.0938217425411 -37.82680668646078, 145.093757 | 0 | 5 | 12 | 27 |
| 4 | 20077970000 | MULTIPOLYGON (((144.79178225632853 -37.72146450518837, 144.79221 | 0 | 2 | 6 | 12 |
| 5 | 20089930000 | MULTIPOLYGON (((144.8170857255147 -37.781313267272544, 144.81829 | 0 | 3 | 8 | 15 |
| 6 | 20123270000 | MULTIPOLYGON (((145.2624318640609 -38.00022221920827, 145.262432 | 0 | 1 | 7 | 14 |
| 7 | 20128350000 | MULTIPOLYGON (((145.34070775627814 -38.21564233599613, 145.33909 | 0 | 0 | 0 | 3 |
| 8 | 20196590000 | MULTIPOLYGON (((145.05876665697943 -37.933250114707626, 145.0566 | 1 | 3 | 9 | 19 |
| 9 | 20230610000 | MULTIPOLYGON (((145.14268188721383 -37.979895140779426, 145.1428 | 0 | 2 | 8 | 15 |
| 10 | 20280240000 | MULTIPOLYGON (((144.75659347562907 -37.878113303780665, 144.7562 | 1 | 1 | 5 | 7 |
| 11 | 20334050000 | MULTIPOLYGON (((145.2299805067229 -37.846648796001325, 145.22997 | 0 | 4 | 9 | 20 |
| 12 | 20335760000 | MULTIPOLYGON (((145.2405273650536 -37.84518781672491, 145.240511 | 0 | 5 | 9 | 18 |
| 13 | 20425540000 | MULTIPOLYGON (((145.07813136684373 -37.903708435856906, 145.0779 | 0 | 2 | 12 | 23 |
| 14 | 20443000000 | MULTIPOLYGON (((145.146271087207 -37.88348875043088, 145.1461593 | 0 | 2 | 12 | 25 |
| 15 | 20474250000 | MULTIPOLYGON (((144.91150219654452 -37.71271140326172, 144.91086 | 1 | 4 | 11 | 22 |
| 16 | 20476110000 | MULTIPOLYGON (((144.97137343651175 -37.709710637329714, 144.9714 | 0 | 4 | 13 | 24 |
| 17 | 20479971000 | MULTIPOLYGON (((145.19054889150982 -38.316023366160245, 145.1905 | 0 | 1 | 1 | 2 |
| 18 | 20493950000 | MULTIPOLYGON (((144.82256689973863 -38.378797831318835, 144.8226 | 0 | 0 | 2 | 2 |
| 19 | 20495750000 | MULTIPOLYGON (((145.05467769396523 -38.4202704474192, 145.056188 | 0 | 0 | 0 | 1 |
| 20 | 20503520000 | MULTIPOLYGON (((145.04158987641446 -38.23847987540726, 145.04183 | 1 | 2 | 2 | 5 |
| 21 | 20503810000 | MULTIPOLYGON (((145.0967185244693 -38.168231198768744, 145.09424 | 0 | 1 | 6 | 7 |
| 22 | 20544680000 | MULTIPOLYGON (((144.9886445598667 -37.8354269392916, 144.9885990 | 0 | 5 | 12 | 22 |
| 23 | 20595900000 | MULTIPOLYGON (((145.19172424541674 -37.82801972323206, 145.19363 | 0 | 4 | 11 | 19 |
| 24 | 20604780000 | MULTIPOLYGON (((145.14274490134613 -37.85141618998845, 145.14247 | 0 | 5 | 11 | 25 |

**Figure 10.** The table in a database which contains the number of POIs buffer that overlaid with mesh blocks.

Weight Calculation and Query Point Classification Method

The weight attributes, securely stored in the database, play a crucial role in QGIS. They are essential for calculating the density of POIs and creating a heatmap for each query point. Values from overlapping POI buffer zones at various distances are combined to ensure a comprehensive representation. This combined value is a crucial parameter in the QGIS value function, which is necessary for creating both the heatmap and graduated visualizations. The decision to combine all POI buffer weights is made to provide a clear and consistent representation of the overall POI density. This aggregated weight, derived from the intersect count, is also used with the data classification in QGIS, thus contributing to the systematic classification of query points based on their weights.

In this research, quantile classification was employed to classify query points. This approach offers several advantages, including its capacity to distribute query points uniformly across different classes, thereby providing a balanced representation. Additionally, quantile classification is less sensitive to outliers, thereby ensuring that extreme values do not disproportionately influence the classification results. This robustness makes it particularly suitable for scenarios where the distribution of POI density exhibits variations, thereby allowing for a more reliable and resilient classification process. Considering future research directions, exploring alternative heatmap calculation methods, such as kernel density estimation (KDE), may be valuable. Experimenting with different techniques could offer insights into potential variations in heatmap results. Moreover, alternative data classification methods beyond quantile classification could be explored in future investigations, thereby enhancing the methodology's adaptability and addressing potential variations in data distribution. Discussing these possibilities in the future work section can highlight potential enhancements to the methods and acknowledge the continuous improvement and refinement of research techniques.

3.3.2. Classification Process

To ensure comprehensive and accurate results, 30 query points were randomly selected from the residential address dataset in each city under study, thereby preventing clustering in a single location. This approach encourages a diverse representation of scenarios within the sample, thus enhancing result variability. The categorization of each query point into distinct groups based on the density of the surrounding POIs is a pivotal step in the methodology. This categorization employs four varying buffer sizes to calculate density, thereby assigning each point to the appropriate category.

The density calculation involves overlaying the buffer on the mesh block, thereby incorporating weights derived from overlapping POI buffer zones at different distances. The resulting values generate a heatmap (see Figures 11–15), thus categorizing density into five distinct classes. Each class represents a unique density level, thus facilitating the observation and analysis of different scenarios. This classification process aids in evaluating the accuracy of Euclidean and road network distance methods under varying POI density conditions.

Significantly, the characteristics of POI density varied between datasets, especially in the ambulance and hospital datasets. Despite utilizing the same query points in each dataset, the categorization may differ due to these density variations. The experiment comprised 300 samples, with 30 query points randomly selected from each city in the hospital and ambulance datasets. The cumulative count of query points in each category is as follows: Class One (73 points), Class Two (58 points), Class Three (55 points), Class Four (55 points), and Class Five (59 points). This ensures a reasonably equitable distribution, thereby minimizing the impact of data imbalance on the results.

The balanced distribution of samples across categories enhances the reliability of experimental results, thereby allowing for a robust evaluation of the accuracy of both Euclidean and road network distance methods. This balanced approach minimizes potential biases from unequal representations of different density levels, thus enabling a fair comparison

between distance methods in diverse scenarios. The number of query points in each class is summarized in Table 2, thus further substantiating the experimental outcomes.

**Table 2.** Number of query points in each class for each city.

| Class | Adelaide | Brisbane | Melbourne | Perth | Sydney | Total Number of Points |
|---|---|---|---|---|---|---|
| One | 15 | 18 | 13 | 15 | 12 | 73 |
| Two | 10 | 7 | 12 | 15 | 14 | 58 |
| Three | 13 | 14 | 11 | 7 | 10 | 55 |
| Four | 10 | 9 | 12 | 12 | 12 | 55 |
| Five | 12 | 12 | 12 | 11 | 12 | 59 |

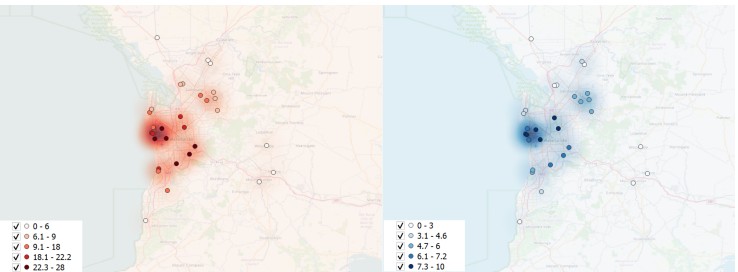

**Figure 11.** Left heatmap and query points from ambulance dataset; right heatmap and query points from hospital dataset in Adelaide.

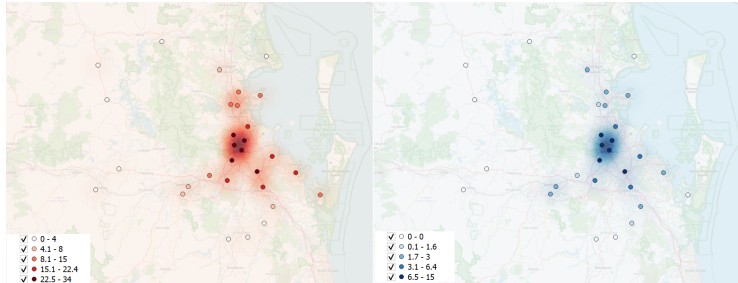

**Figure 12.** Left heatmap and query points from ambulance dataset; right heatmap and query points from hospital dataset in Brisbane.

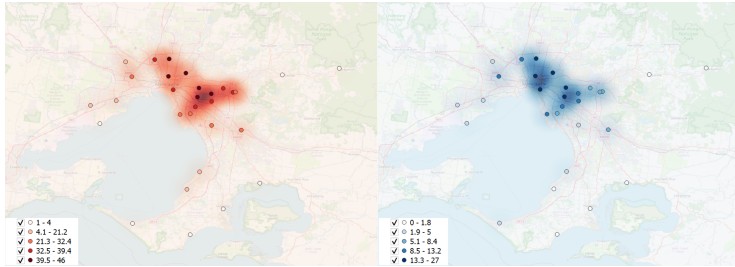

**Figure 13.** Left heatmap and query points from ambulance dataset; right heatmap and query points from hospital dataset in Melbourne.

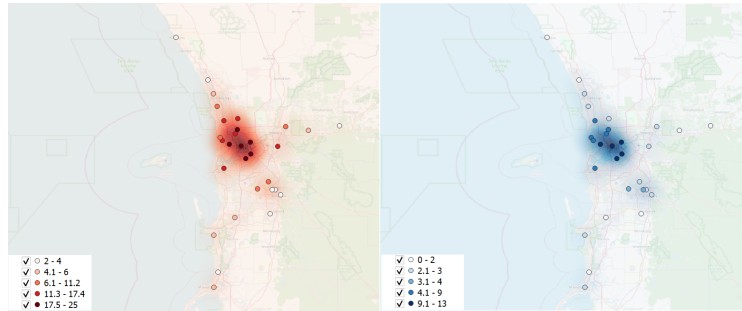

**Figure 14.** Left heatmap and query points from ambulance dataset; right heatmap and query points from hospital dataset in Perth.

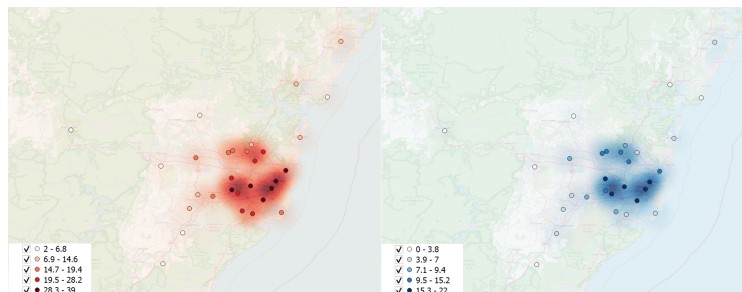

**Figure 15.** Left heatmap and query points from ambulance dataset; right heatmap and query points from hospital dataset in Sydney.

*3.4. Euclidean Search*

To obtain results using the Euclidean distance method, distances were computed based on the query point's latitude and longitude and the POIs stored within the spatial database, thereby measuring the straight-line distance between them. These calculated distances were then organized in a dedicated column within the database. Subsequently, by employing the precomputed Euclidean distances, both the kNN and range searches were conducted on the spatial databases. This involved implementing kNN queries to find the nearest neighbors and range queries to identify POIs falling within specific distance thresholds. The detailed process of obtaining results for both query methods provides an understanding of the research methodology.

3.4.1. K-Nearest Neighbors Search in Euclidean Distance

The kNN queries method searched for the nearest POIs to the query point based on the number of points specified by the user, which was set to be limited to the first 20 data points (k = [1, 20]). These points were sorted based on their distances from the query point. The distances between query points and POIs were calculated on the database using the SQL statement presented in Listing 1.

**Listing 1.** SQL statement for calculate Euclidean distance.

```
SELECT
ST_SETSRID(ST_MAKEPOINT(olon,olat),4326)::geography<->geom::geography
AS dist_eu
FROM hospital;
```

In this statement, the "geom" column contains the geographical coordinates of individual hospitals, while "olat" and "olon" denote the latitude and longitude of the designated query point, respectively. Following the distance calculation, the results were ordered according to distance in ascending order, and only the top 20 data points were selected using the SQL statement shown in Listing 2.

**Listing 2.** SQL statement for retrieving kNN result for Euclidean distance method.

```
SELECT *,
ST_SETSRID(ST_MAKEPOINT(olon,olat),4326)::geography<->geom::geography
AS dist_eu
FROM hospital_q1
ORDER BY dist
LIMIT 20;
```

The retrieved data from this SQL statement represents the kNN search results for values of k between 1 and 20. For instance, the first row corresponds to the result of 1NN, and the first to fifth rows collectively represent the result of 5NN.

3.4.2. Range Search in Euclidean Distance

Unlike the kNN query method, the range query method searched for POIs within a specific distance range determined by the user. This method retrieves all POIs located within the specified range, and the results are also sorted based on their distance from the query point. Hence, the number of results is not limited. This study employed distance thresholds of 1 to 150 km in 5 km increments for the range search method.

For the range search, measuring distances between query points and POIs is the same as the kNN search method. However, the key distinction lies in the result limitation, which is based on distance rather than a set number of rows. Consequently, the SQL statement conditions are adjusted to limit the distance rather than restricting the number of rows. The SQL statement in Listing 3 is provided for retrieving results within a 5 km range.

**Listing 3.** SQL statement for retrieving range query results within a 5 km for Euclidean distance.

```
SELECT *
FROM (SELECT *,
ST_SETSRID(ST_MAKEPOINT(olon,olat),4326)::geography<->geom::geography
    dist_eu
FROM hospital_q1)
WHERE dist_eu < 5
ORDER BY dist_eu;
```

*3.5. Road Network Search*

The distance between query points and POIs is calculated using the road network distance extracted from the Open Source Routing Machine (OSRM) API. OSRM is an open-source software designed for rapid route calculations, thereby making it well suited for real-time applications and scenarios involving large-scale road networks. It provides accurate and efficient routing results, thereby delivering the shortest route in terms of distance and duration time. The OSRM project is a collaborative effort, and the protocol implemented by the service is version 1 for all OSRM 5.x installations. As of the latest implementation, we are using OSRM version 5.27.1 (Open Source Routing Machine, Dennis Luxen, http://www.project-osrm.org, accessed on 7 March 2023).

In this study, only the distance obtained from OSRM was utilized to derive results. After extracting distances from OSRM, kNN and range queries were performed to obtain results for each method.

3.5.1. K-Nearest Neighbors Search in Road Network Distance

The process of obtaining results through the kNN search using the road network distance method closely aligns with the Euclidean distance method. However, a key difference lies in the approach. While the Euclidean distance method calculates the direct distance between a query point and points of interest using a spatial database, the road network distance method acquires distances by utilizing the Open Source Routing Machine (OSRM) API through Python code. This API extracts the shortest route, and the resulting distances are then stored in a specified database table.

When conducting kNN queries using the road network distance method, data are sorted in ascending order based on the distances in the osrm_dist column. Following this, the results are constrained to the first 20 data points, thereby aligning with the range of k values from 1 to 20. This process reflects the Euclidean search method, where a corresponding SQL query is employed, as shown in Listing 4.

**Listing 4.** SQL statement for retrieving kNN results for the road network distance method.

```
1  SELECT *
2  FROM hospital_q1
3  ORDER BY osrm_dist
4  LIMIT 20;
```

In this context, osrm_dist symbolizes the column containing distances measured along the shortest routes on genuine road networks between the query point and POIs. The constraint to the initial 20 data points is influenced by the range of k values, thereby providing a comprehensive set of potential nearest neighbors for the given query.

### 3.5.2. Range Search in Road Network Distance

In the road network distance method, the objective of range queries was to retrieve results within specific distance intervals utilizing actual road distance. The distance thresholds ranged from 1 km to 150 km, with increments of 1 km, 5 km, and subsequent 5 km intervals (1, 5, 10, 15, 20, 25, 30, . . . , 150). The process involved extracting distances from the Open Source Routing Machine (OSRM) and executing SQL queries to limit the distance for each interval. An illustrative example of a SQL statement for retrieving results within a 5 km interval is shown in Listing 5.

**Listing 5.** SQL statement for retrieving range query results within a 5 km for road network distance.

```
1  SELECT *
2  FROM hospital_q1
3  WHERE osrm_dist < 5
4  ORDER BY osrm_dist;
```

This SQL statement demonstrates the retrieval process for the 5 km interval. Similar queries were executed for each distance threshold, thereby encompassing the 1 km to 150 km range. The osrm_dist column represents distances measured along the road network between the query point and POIs.

### 3.6. Accuracy Calculation

#### 3.6.1. K-Nearest Neighbor Queries Accuracy Calculation

To calculate the accuracy of the Euclidean distance method for each category of query point and each number of k in the kNN search method, the following formula was used:

$$\text{Accuracy} = \left( \frac{\text{Total number of correct results}}{\text{Total number of retrieved results}} \right) \times 100\% \qquad (1)$$

The correct results were defined by comparing the set of POIs obtained from the Euclidean distance method with the collection of POIs obtained from the road network distance method. If the set of POIs obtained from the Euclidean distance method is different in any way from the set obtained from the road network distance method, the result is marked as incorrect.

#### 3.6.2. Range Queries Accuracy Calculation

For the range query, this method retrieved all POIs within the specified distance range, which was set to 5, 10, 15, 20, 25, 30, 35, . . . , and 150 km, and the results were sorted based on their distance from the query point. As the number of retrieved POIs was not limited in the range search method, the percentage of accuracy was calculated by following these steps:

Number of Incorrect results (Euclidean) = Total number of points retrieved from the Euclidean distance method − Total number of points retrieved from the road network distance method

Number of Correct results (Euclidean) = Total number of points retrieved from the Euclidean distance method − Number of Incorrect results (Euclidean)

Accuracy = (Number of Correct Results (Euclidean)/Total number of points retrieved from the Euclidean distance method) × 100%

## 4. Results

### 4.1. Comparison of k-Nearest Nighbors Search

This study compared the Euclidean distance method and the road network distance method for each category. The comparison was conducted by retrieving the POIs using both methods and comparing the results. For the kNN queries, if the POIs retrieved from both methods were identical, the result would be marked as 1; otherwise, it was marked as 0. This approach was applied to each classification, thereby allowing us to compute accuracy using the following formula:

$$\text{Accuracy}_{k,c} = \frac{\sum_{i=1}^{n} \text{kNN}_c \times 100\%}{\text{total cases}_c} \tag{2}$$

where:

$$k = \text{the number of k-nearest neighbors}$$
$$c = \text{density category (1–5)}$$

Each category's accuracy results were plotted and presented in Figure 16. It is interesting to observe that as the value of k increased, the classification accuracy decreased across all categories. Even for query points with the highest density of POIs, the accuracy still decreased. The results indicate that the accuracy remained high for Class One and Three, with rates above 85%, as well as for the remaining classes, which had rates exceeding 90%. This suggests that the Euclidean and road network distance methods consistently provide accurate results, particularly for the first category of points of interest.

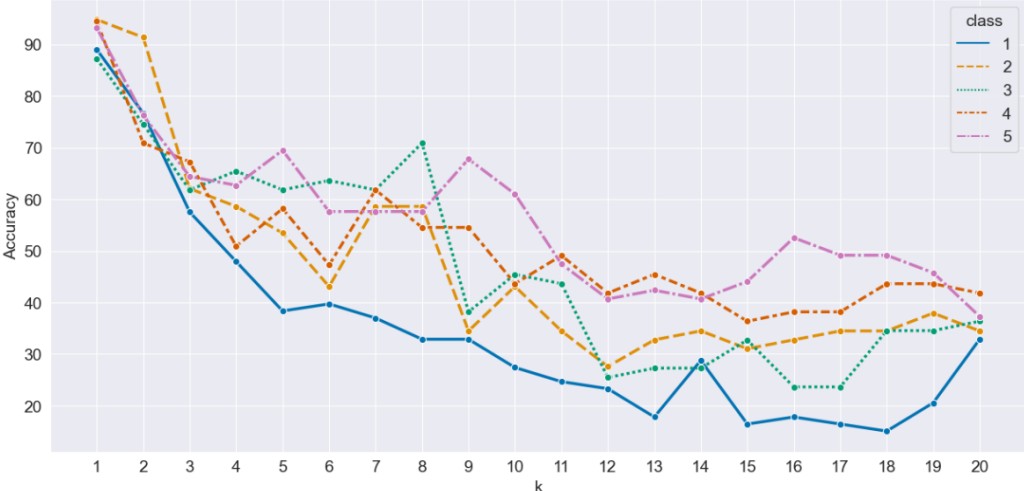

**Figure 16.** Accuracy of the Euclidean distance method using k-nearest neighbor queries for all samples.

The accuracy significantly dropped for all classes except for Class Two when the number of k was set to two, thus maintaining an accuracy rate above 90%. Nevertheless, the accuracy rate for these classes remained above 70%. It appears that increasing the number of nearest neighbors beyond two further decreased the accuracy for all classes, with the accuracy dropping below 70% across the board.

As depicted in Figure 17, the box plot offers a thorough overview of accuracy distribution while drawing attention to any data points that fall outside the typical range (outliers). It is clear that as the value of k exceeded two, the median accuracy dropped below 70% across all categories. This implies that utilizing more than two nearest neighbors results in a noticeable reduction in classification accuracy. Additionally, the presence of outliers beyond the whiskers in specific categories points to situations where the accuracy performance becomes notably poor or erratic.

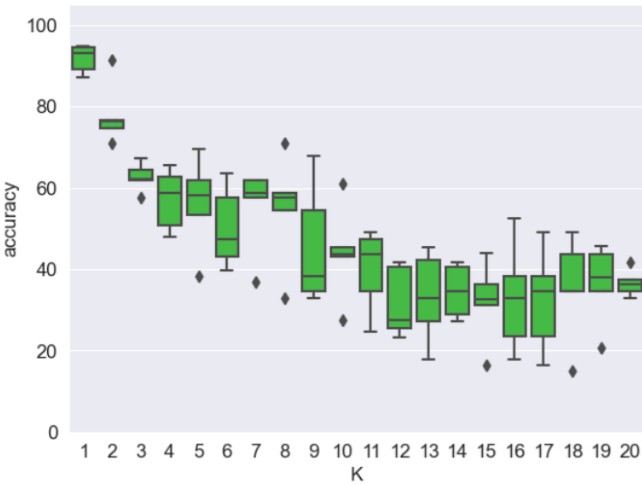

**Figure 17.** Accuracy of the Euclidean distance method using k-nearest neighbor queries for all samples and all classes.

Analyzing Figures 16 and 17, we observe the results obtained from queries in both types of POIs. Considering the differences between two different dataset types is crucial, since the number of POIs significantly differs. The hospital dataset contains fewer data points than the ambulance dataset. Hence, the accuracy results for each value of k are expected to differ between the two datasets. By taking into account these differences, we can develop a more comprehensive insight into how dataset attributes influence the accuracy of classification outcomes. This examination enables us to uncover the nuances that distinguish the hospital dataset from the ambulance dataset, thereby offering valuable insights for future enhancements and similar analyses.

Figure 18 shows line graphs that plot the accuracy separately for the hospital, ambulance, and combined datasets. Each class is represented in its subplot. The purpose of the graph is to observe the accuracy of each dataset of the Euclidean distance method result compared with the benchmark as a result of the road network distance method.

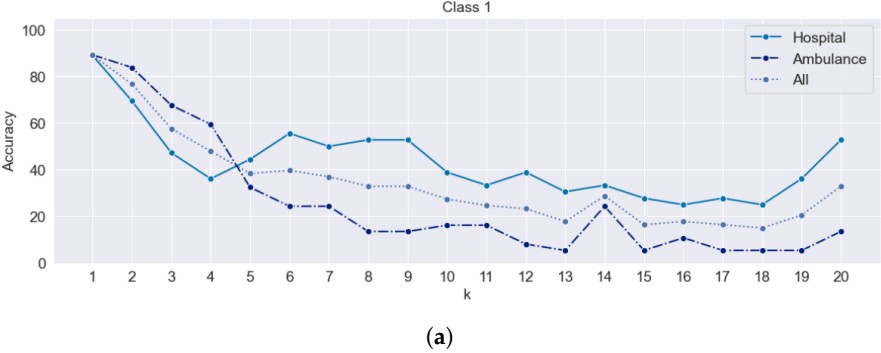

(**a**)

**Figure 18.** *Cont.*

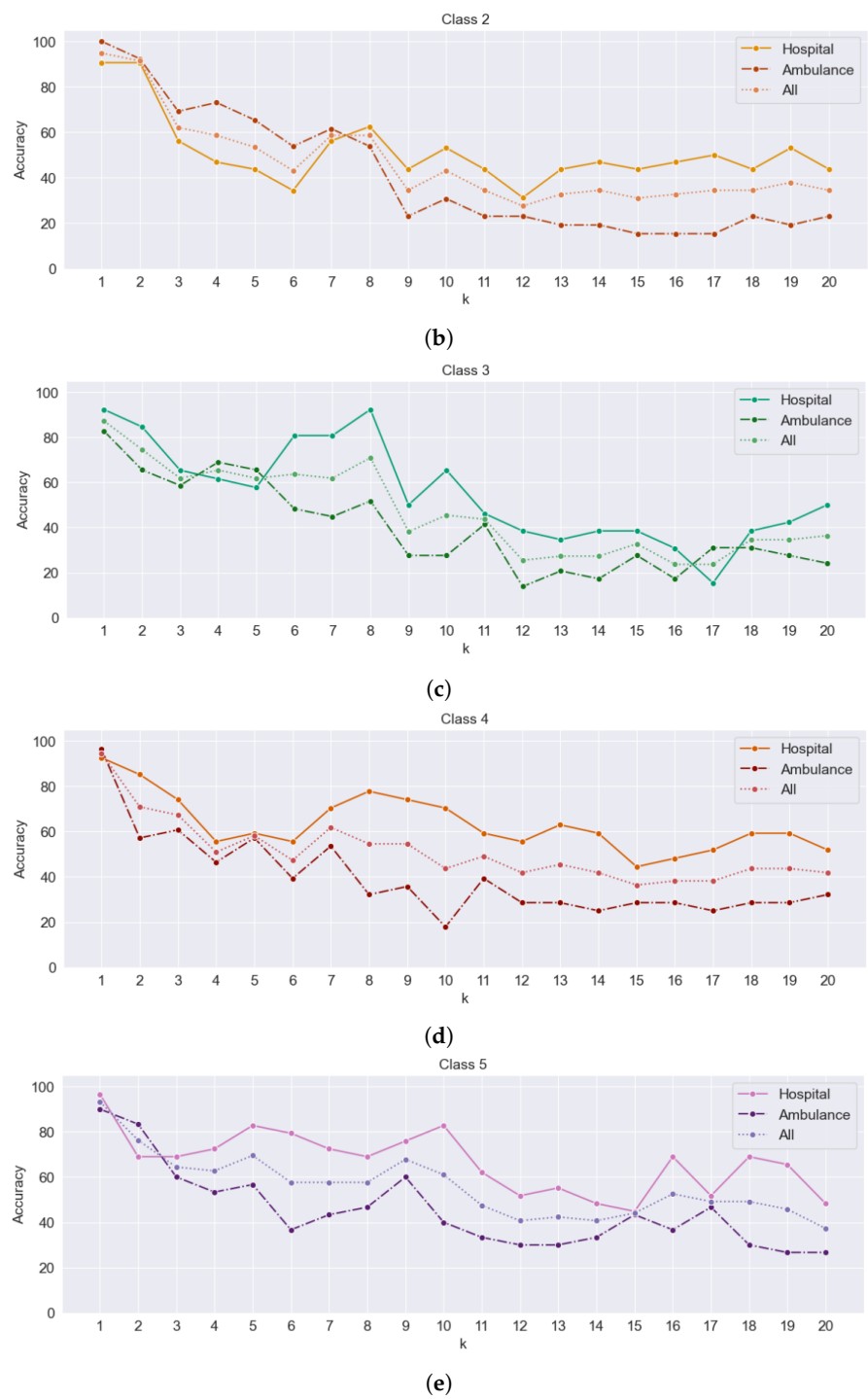

**Figure 18.** Line graph plot with the different datasets (hospital, ambulance, and hospital and ambulance) for kNN queries. (**a**) Class 1—line graph plot with the different datasets. (**b**) Class 2—line graph plot with the different datasets. (**c**) Class 3—line graph plot with the different datasets. (**d**) Class 4—line graph plot with the different datasets. (**e**) Class 5—line graph plot with the different datasets.

The observation from the graph is that as the value of k increased, the hospital dataset, which has a smaller number of POIs than the ambulance dataset, tended to achieve higher accuracy results. On the other hand, the ambulance dataset, with its large number of POIs, exhibited lower accuracy results. This suggests that having a smaller number of POIs in the dataset leads to higher accuracy in some classes and some values of k-nearest neighbors when using the Euclidean distance method.

### 4.2. Comparison of Range Search

The accuracy for the Euclidean distance method using range queries was calculated based on the number of POIs retrieved from both methods, with the number of results from the road network distance method serving as a benchmark. The following formula was applied to calculate the accuracy:

$$\text{Incorrect Eu} = \text{Total Eu} - \text{Total Rd}$$
$$\text{Correct Eu} = \text{Total Eu} - \text{Incorrect Eu}$$
$$\text{Accuracy} = \frac{\text{Correct Eu} \times 100\%}{\text{Total Eu}}$$

where:

$$\text{Eu} = \text{The number of results from the Euclidean method}$$
$$\text{Rd} = \text{The number of results from the road network distance}$$

Since the Euclidean distance method distances are equal to or shorter than the road network distance method, we can subtract the total number of results from the road network distance method from the total number of results from the Euclidean distance method to determine the number of incorrect results. This approach allows us to assess the accuracy of the Euclidean distance method in retrieving spatial data through range queries compared to the benchmark provided by the road network distance method.

The accuracy of the range queries was determined for each query point within a range of distances, including 1 km, 5 km, 10 km, 15 km, 20 km, and subsequent increments up to 150 km. We retrieved the sets of results for each specific distance range from the Euclidean and the road network distance method. Subsequently, we calculated the accuracy using the abovementioned formula, thereby considering the total number of results obtained from each method. These accuracy results were then utilized to generate the graph presented in Figure 19, which visually represents the accuracy of different values of k within each class for the respective range distances. The graph indicates that only two categories exhibited accuracy results for the 1 km range. The shaded region on the line graph represents the confidence interval, which primarily lies above a 50% accuracy level. Due to the limited number of cases within the 1 km range, no shading is present on the line corresponding to this range. Despite Class One having the lowest point of interest density, some cases demonstrated accuracy values comparable to the road network distance method within the 1 km range. Conversely, Class Five, which exhibited the highest point of interest density, exhibited lower accuracy than Class One across the 1 km, 5 km, 10 km, and 15 km ranges. Furthermore, the graph analysis reveals intriguing patterns regarding the relationship between range and accuracy across different classes. Notably, all classes exhibited similar trends: as the range increased, the accuracy initially decreased. However, as the range continued to increase, an interesting pattern emerged. At a certain point, the accuracy began to consistently and significantly improve for each class, thereby indicating the presence of an optimal range beyond which accuracy starts to rise despite the increasing range. The turning point at which accuracy started to improve varied across classes, with Class Five exhibiting the earliest improvement at a relatively low range, followed by Class Four at a slightly higher range, and so forth.

The accuracy analysis was conducted to compare the performance of the hospital and ambulance datasets for each range and class. The accuracy difference between the two datasets was plotted, following a similar process as the one used for kNN queries, as shown in Figure 20. Interestingly, the hospital dataset exhibited higher accuracy than the ambulance dataset, despite the latter having a higher number of POIs. In Class One, for the range queries within 1 km and 5 km, there were no POIs within this range in the hospital dataset. Similarly, in Class Five, only the ambulance dataset contained data within the 1 km range. However, aside from these specific cases, the hospital dataset consistently demonstrated higher accuracy across all ranges and classes. Additionally, it is essential

to note that the shading area representing the accuracy for all ranges and classes in each dataset consistently remained above the 50% threshold. However, a notable exception exists in Class One, where the shaded region fell below 50% at specific ranges. This discrepancy suggests that the accuracy of the range queries within those specific ranges was relatively low for Class One. This observation concludes that the range queries' accuracy values were generally not uniformly high across all classes, as indicated by the shading area only above 50% accuracy.

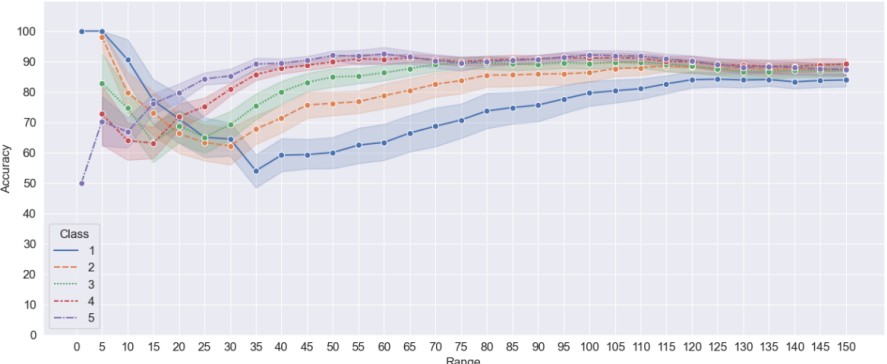

**Figure 19.** Accuracy of the Euclidean distance method with range queries all samples.

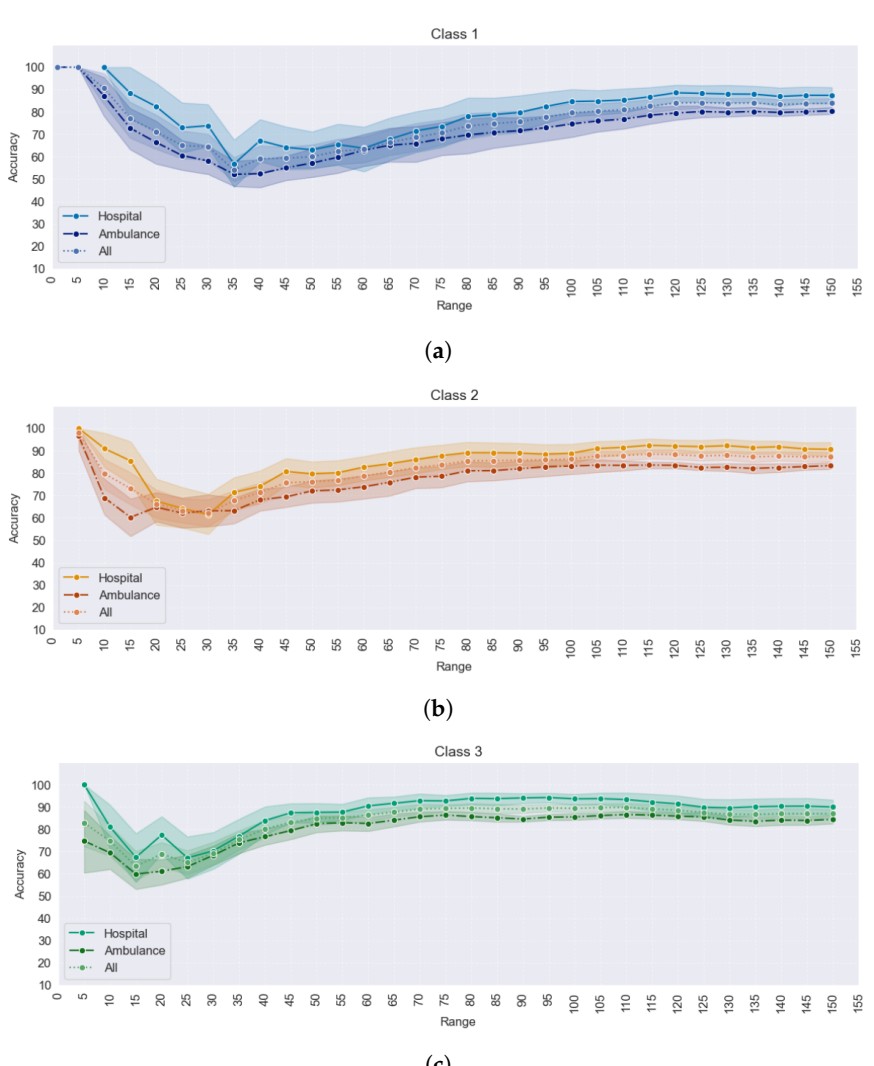

**Figure 20.** *Cont.*

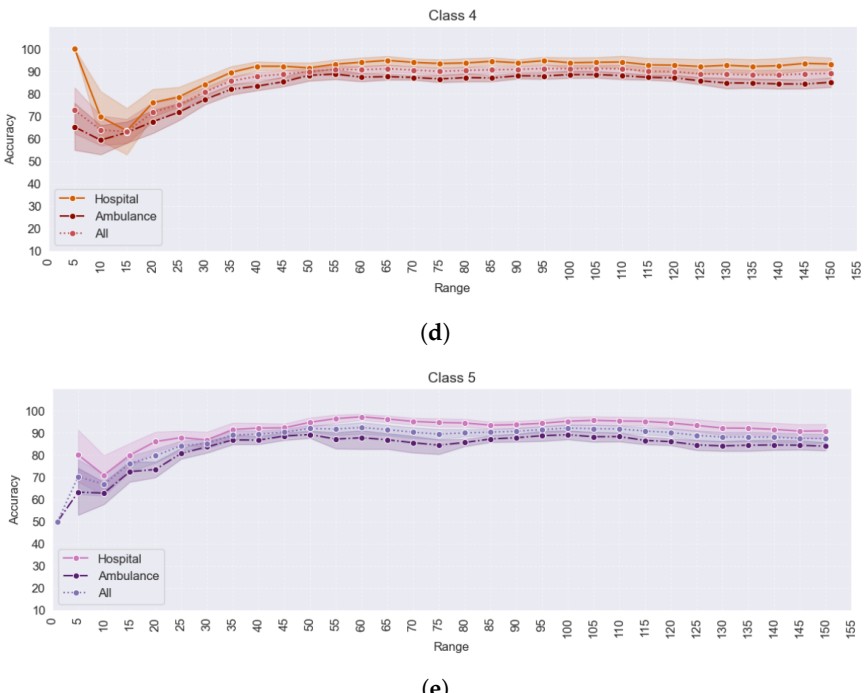

(d)

(e)

**Figure 20.** Line graph plot with the different datasets (hospital, ambulance, and hospital and ambulance for range queries). (**a**) Class 1—line graph plot with the different datasets. (**b**) Class 2—line graph plot with the different datasets. (**c**) Class 3—line graph plot with the different datasets. (**d**) Class 4—line graph plot with the different datasets. (**e**) Class 5—line graph plot with the different datasets.

## 5. Discussion

The primary objective of this research was to evaluate the accuracy of the Euclidean distance method compared to the road network distance method. The Open Source Routing Machine (OSRM) was utilized to extract the road network distances, which served as the benchmark for comparison with the Euclidean distance method. OSRM provided a reliable and efficient means of calculating the actual distances based on the road network topology. By using OSRM as the benchmark, we were able to assess the accuracy of the Euclidean distance method in capturing the actual distances along the road network. Furthermore, the study aimed to comprehensively examine the unique characteristics associated with each classification of query points and to rigorously evaluate the accuracy of each category. Our analysis disclosed the accuracy rates of each query method when measured against the Euclidean and road network distances, with the latter serving as the established benchmark. In kNN queries, we observed a correlation between the accuracy and the number of k. Specifically, we found that as the number of k increased, the accuracy for each class of query points consistently decreased. This observation could be attributed to the intricate road structure within the city. As the value of k rises, the algorithm considers a higher number of nearby points, thus potentially leading to a greater diversity in road structures and terrain. This diversity might introduce variations in the distances calculated, thereby impacting the accuracy of the Euclidean distance method. The road network's complexity, including factors like one-way streets, intersections, and irregularities, could also contribute to this phenomenon. This finding highlights the impact of the chosen value of k on the accuracy of the Euclidean distance method. Regarding the range query, we identified a consistent pattern across different classes, with each class exhibiting a similar trend except for the turning point in the range value. The turning point, which represents the threshold where the accuracy starts to increase, is influenced by the density of the POIs. Class Five showed the earliest increase in accuracy, followed by Class Four, thus suggesting a relationship between the class and the point at which accuracy begins to improve.

However, the increased accuracy in this context may suggest that the distance range adequately covers all surrounding points of interest (POIs). As previously mentioned in the kNN case, where road structure influenced the decrease in accuracy, we saw a correlated phenomenon here. The high-density class experienced a reduction in accuracy when the range distance was less, thus indicating a potential interplay between road structure, density, and range distance that impacts accuracy trends. Another insight gained from this research is that many POIs do not necessarily guarantee high accuracy. Despite the hospital dataset having fewer POIs than the ambulance dataset, the accuracy of the hospital dataset was found to be higher. This observation suggests that when there are limited choices (few POIs), the retrieved results using the Euclidean distance method may closely align with those obtained from the road network. Nevertheless, as the number of available choices increases, there is a likelihood of encountering slight differences in distances, thereby potentially influencing the final results and impacting the overall accuracy. This significant finding has the potential to contribute to future research endeavors substantially focused on analyzing the conditions under which the Euclidean distance can be employed as a viable trade-off between accuracy and computational time, particularly when considering the classification of query points based on POI density. By highlighting the circumstances in which this alternative metric performs acceptably across different density classes, our study provides valuable insights for practical implementations. This understanding becomes particularly crucial for scenarios where the computational efficiency gained from using Euclidean distance outweighs the marginal decrease in accuracy, thereby offering a practical solution for certain types of queries in real-world applications.

Previous studies have extensively investigated the Euclidean distance method's accuracy and used the road network distance method as a benchmark. Notably, Hua, Xie, and Tanin (2018) [29] employed Euclidean distance as a computationally efficient alternative for estimating road network distance, specifically in group nearest neighbor (GNN) queries. The experiments compared the Euclidean decider, which utilizes Euclidean distance, with road network distance as a benchmark. Real road network graphs from various cities created scenarios with different query points and POI distributions. The study varied parameters such as the number of query points, POI density, value of k, and road network structure. The results indicate that the Euclidean metric achieved high accuracy levels, exceeding 74.7% for kNN queries and ranging from 86% to 95% for general GNN queries, thereby showcasing its effectiveness in achieving a balance between accuracy and computational efficiency in transportation services.

In comparison, our studies demonstrated high accuracy at certain values of k when using kNN queries and achieved an accuracy rate above 50% for range queries. The results from their approach for kNN queries showed significantly higher accuracy. This difference may arise due to the influence of various parameters or variables affecting accuracy. The substantial contrast in outcomes could be attributed to specific conditions, configurations, or factors inherent in their approach compared to ours. Further investigation into the particular variables or parameters contributing to this marked variation in accuracy would be essential for a comprehensive understanding of the results.

Our study categorized the query points by utilizing the attribute weight derived from counting the number of POI buffers overlaid with the mesh block containing the query points. This approach enabled us to use the weight attribute to calculate the density using a heatmap. However, it is important to acknowledge several limitations that may impact the accuracy of the result. Firstly, the query point classification criteria, including the weight attribute selection, the method employed for heatmap calculation, and geotypes, can introduce variability into the results. Additionally, the road structure for each city potentially affects the result of the POIs for the road network distance method, which might impact the accuracy of the Euclidean distance. Furthermore, we recognize that there may be an imbalance in the number of POIs across different cities. This could potentially introduce bias in the analysis, as certain cities may have a higher concentration of POIs

than others. This imbalance could influence the results, particularly when comparing the accuracy of the Euclidean distance method across different cities.

By acknowledging these limitations, we demonstrate a critical awareness of the potential factors that could influence the accuracy and reliability of our study's results.

## 6. Conclusions

In conclusion, our study aimed to evaluate the accuracy of the Euclidean distance method compared to the road network distance method using kNN queries and range queries, thus considering the classification of query points. The investigation brought several significant insights.

Our findings unveiled several critical insights. Specifically, in kNN queries, peak accuracy was achieved when k was set to one, thereby exceeding 85% across all classes. However, with an increase in k, a consistent decrease in accuracy was observed, thereby highlighting the sensitivity of results to the choice of k. Turning to range queries, a uniform accuracy pattern emerged across classes, with the turning point influenced by the varying range values. Notably, a higher density of POIs correlated with an earlier increase in accuracy.

Furthermore, the comparative analysis between ambulance and hospital datasets questioned the assumption that more POIs ensure superior accuracy. Despite containing fewer POIs, the hospital dataset demonstrated higher accuracy, thus emphasizing the complex interplay between dataset characteristics and method performance. Our study contributes substantially by illuminating scenarios in which the Euclidean distance method is a feasible compromise between accuracy and computational efficiency, particularly in classifying query points based on POI density.

The discussion provides a detailed understanding of the relationship between accuracy and the number of k in kNN queries. As k increased, the diversity in road structures and terrain introduced variations, thus impacting accuracy. Similarly, the observed increase in accuracy in range queries suggests that the distance range adequately covered surrounding POIs. However, the study cautions about potential interactions between road structure, POI density, and range distance influencing accuracy trends.

While our approach adds valuable insights to the existing body of research, we acknowledge certain limitations. The query point classification criteria, the weight attribute selection, the heatmap calculation method, and the consideration of different geotypes introduce variability into our results. The impact of road structure on the accuracy of POIs in the road network distance method and potential bias due to an imbalance in the number of POIs across cities could influence the accuracy comparison of the Euclidean distance method.

In future work, accounting for road structure in query point classification and addressing POI imbalances across cities will refine our understanding of the Euclidean distance method's performance. Additionally, exploring the impact of criteria for classification, the number of samples, the road network algorithm for extracting the shortest route, the weight attribute for heatmap calculation, and different geotypes (such as dense urban areas and sparse rural areas) will provide a more comprehensive assessment. By doing so, we aim to enhance the applicability of our findings to real-world scenarios while mitigating potential biases and improving the robustness of our methodology.

**Author Contributions:** Conceptualization, K.A. and P.T.; methodology, D.T., K.A. and PT.; software, D.T. and K.A.; validation, D.T., K.A. and P.T.; formal analysis, P.T.; investigation, P.T.; resources, K.A.; data curation, K.A.; writing—original draft preparation, P.T.; writing—review and editing, D.T., K.A. and P.T.; visualization, P.T.; supervision, D.T. and K.A.; project administration, D.T. and K.A. All authors have read and agreed to the published version of the manuscript.

**Funding:** This research received no external funding.

**Data Availability Statement:** No new data were created or analyzed in this study. Data sharing is not applicable to this article.

**Conflicts of Interest:** The authors declare no conflicts of interest.

## Abbreviations

The following abbreviations are used in this manuscript:

| | |
|---|---|
| kNN | k-Nearest Neighbors |
| POIs | Points of interest |
| OSM | OpenStreetMap |
| OSRM | Open Source Routing Machine |
| G-NAF | Geocoded National Address File |
| ASGS | Australian Statistical Geography Standard |

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
