# Peer review of "Navigating the Maps: Euclidean vs. Road Network Distances in Spatial Queries"

_algorithms, doi:10.3390/a17010029_

Round 1
Reviewer 1 Report
Comments and Suggestions for Authors
Algorithms-2757827: Navigating the Maps: Euclidean vs. Road Network Distances in Spatial Queries
The paper introduces a comparative study on the accuracy of retrieved POIs with the k nearest neigbours and range search methods based on road distance and Euclidean distance
Although the paper addresses the problem of efficiently retrieving POIs from spatial databases, I have several concerns
My main issue is the writing. I had a hard time to follow the main line of argumentation, since the paper lacks conciseness. Many parts are repetitive, which makes it hard for the reader to focus on the core of the paper. The main message gets diluted. I am strongly convinced that the length of the paper could be reduced way below 20 pages.
Examples of repetitive passages are section 2.2 and 2.2.1; section 3 and 3.1; section 3.2.; section 3.3.1; and section 6.
Another issue with language, is the mixing of terms, such as “k”, “class”, and “category”, often referring to either different or the same things throughout the paper which is quite confusing for the reader.
Regarding related work (section 2.3) I wonder if the two referenced studies are all? I suggest going bit deeper with your literature review. For instance, I suggest to also discuss concepts, such as quadtrees for indexing data in spatial databases and making retrieval more efficient. What exactly is the difference of cited study [12] to yours?
Here are also a few minor suggestions for improvement:
p. 2, line 52: explain E and V
p. 5, line 152: better write “Taniar and Rahyau [4] categorized …”; also briefly explain the mentioned six types.
p. 8: you should explain the concept of Mesh Blocks in more detail. It remains unclear what exactly they are
You never refer to Figure 14 (on p. 9) in the text. Moreover, Figure 14 and Table 1 seem to be highly redundant. I suggest omitting Fig. 14 and renumber all following figures accordingly.
On p. 9, line 275 you mention distances of 1, 5, 10, and 15 km. Why did you choose these distances?
On p. 10 you describe the creation of a heat map based on the Mesh Blocks. How about using KDE? What are the reason for using Mesh Blocks – despite it remains unclear what they exactly are.
The classifications in Figures 18 – 21 have overlapping classes (e.g., 0 – 6, 6 – 9, …). Please adjust the class breaks.
For Figures 25 and 27 you should increase the contrast of the colour hues, so it becomes easier to distinguish the different categories (hospitals, ambulances , all).
I would expect a deeper reflection in the discussion section. What does this all mean? What follows from your results. Can you make any recommendations? In the current state, the paper kind of remains unfinished and lacks a clear contribution.
Some minor language issues are:
p. 7, line 189/190: The verb seems to be missing in this sentence.
p. 7, line 213: in 13 -> in Figure 13
p. 14., line 345: … distance between query points and POIs calculate on the … -> … distance between query points and POIs were calculated on the
p. 15, line 399: closely aligns with the Euclidean distance method. -> This sentence is incomplete; please rephrase.
p. 16, line 452: this step -> these steps
p. 16, line 469: figure 23 -> Figure 23
p. 17, line 489: in this sentence the verb seems to be missing
p. 17, line 492 / p. 18, line 493: Hence, it is expecting the accuracy results for each value of k different between the two datasets. -> Hence, it is expected that the accuracy results for each value of k differ between the two datasets.
p. 18, line 501: … method result compare with the benchmark … -> … method result compared with the benchmark …
Comments on the Quality of English Language
My main issue is the writing. I had a hard time to follow the main line of argumentation, since the paper lacks conciseness. Many parts are repetitive, which makes it hard for the reader to focus on the core of the paper. The main message gets diluted. I am strongly convinced that the length of the paper could be reduced way below 20 pages.
Examples of repetitive passages are section 2.2 and 2.2.1; section 3 and 3.1; section 3.2.; section 3.3.1; and section 6.
Author Response
Dear Reviewer 1:
Thank you for your feedback on ‘Navigating the Maps: Euclidean vs. Road Network Distances in Spatial Queries.’ We have revised the manuscript to incorporate most of the suggestions provided by the Reviewer 1, and these changes are highlighted within the text. Your guidance has been invaluable, and we appreciate the opportunity to enhance the quality of our work based on your insights.
Best regard,
Pornrawee Tatit, Kiki Maulana Adhinugraha, David Taniar
Comment from Reviewer 1:
- Many parts are repetitive, which makes it hard for the reader to focus on the core of the paper. The main message gets diluted. I am strongly convinced that the length of the paper could be reduced way below 20 pages. Examples of repetitive passages are section 2.2 and 2.2.1; section 3 and 3.1; section 3.2.; section 3.3.1; and section 6.
- Thank you for your insightful feedback on the manuscript. We appreciate your attention to detail and the identification of repetitive sections. We acknowledge the concern about the paper's length and share the goal of reducing it to below 20 pages. While we have successfully removed the repetitive parts in sections 2.2, 2.2.1, and restructured section 3 as per your suggestion, we also enriched the related work section with additional research. Consequently, the total length of the research paper still exceeds 20 pages. We apologize for any inconvenience and remain committed to further refining the manuscript to achieve a more optimal length without compromising its substance.
- Regarding related work (section 2.3) I wonder if the two referenced studies are all? I suggest going bit deeper with your literature review. For instance, I suggest to also discuss concepts, such as quadtrees for indexing data in spatial databases and making retrieval more efficient. What exactly is the difference of cited study [12] to yours?
- Thank you for your insightful feedback on the manuscript. We appreciate your attention to detail and the identification of repetitive sections. We acknowledge the concern about the paper's length and share the goal of reducing it to below 20 pages. While we have successfully removed the repetitive parts in sections 2.2, 2.2.1, and restructured section 3 as per your suggestion, we also enriched the related work section with additional research. Consequently, the total length of the research paper still exceeds 20 pages. We apologize for any inconvenience and remain committed to further refining the manuscript to achieve a more optimal length without compromising its substance.
“2. Related work
2.1. Spatial queries in Euclidean distance
Spatial data objects represent a type of data that encompasses multiple dimensions. Over the past few years, there has been a growing utilization of this data type in various applications, including geographic information systems (GIS), computer-aided design, and computer vision [8]. The commonly employed methods for searching spatial objects include k-nearest neighbours (kNN) and range queries techniques.”
“2.1.1. K-nearest neighbour queries in Euclidean distance
The effective execution of Nearest Neighbor (NN) queries, which holds special significance in the field of Geographic Information Systems (GIS) [28], is particularly important when considering the commonly used query type known as k-Nearest Neighbour (kNN). The kNN query involves finding the potential neighbours closest to a given query point [7,28]. In the process of retrieving Points of Interest (POIs) using k-Nearest Neighbours, the distance between the query point and the POIs is calculated by measuring the straight-line distance from the query point’s coordinates to the coordinates of each POI. The resulting POIs are then ordered based on their distance from the query point, enabling the selection of only the top k-ranked POIs from the results. To illustrate, Figure 2 depicts a map diagram with weighted lines representing the actual distance of the road. By employing the Euclidean distance method, the distances are measured as shown in Figure 3. As a result, the retrieved POIs from the Euclidean distance method would be P6, P2, and P7, as indicated in Figure 4.
Fukunaga and Narendra [9] proposed an effective method for computing k-nearest neighbors. Their approach involves a hierarchical decomposition of design samples into disjoint subsets. To achieve this, they applied the branch and bound algorithm, a well-regarded tree-search technique [10, 12, 13]. This algorithm is known for its effectiveness in efficiently searching through the resultant groups. Expanding the scope of k-nearest neighbor (kNN) query processing, Taniar and Rahayu [7] discuss the extension of k-nearest neighbor (kNN) query processing to include obstacles in Euclidean space, known as Obstacle Nearest Neighbors (ONN). ONN involves constructing a visibility graph, representing possible paths around obstacles, and using a disk-based approach with a Restriction method to efficiently process kNN queries while considering obstacle distances. The method iteratively refines search spaces based on obstacle distances to optimize query processing.
Additionally, various types of Nearest Neighbor queries have been studied, including aggregate nearest neighbor (ANN) [3], reverse nearest neighbor (RNN) [27], and group nearest neighbor (GNN) [26]. However, for our specific study, we will focus on the k-nearest neighbor (kNN) queries.”
“2.1.2. Range queries in Euclidean distance
Range queries are an essential query method in database systems [2]. It builds upon the concept of a point query, but in range queries, the area around the query point is expanded, forming an area of query scope. Range queries involve three primary components: points of interest, the query point, and the query scope. To retrieve the relevant objects, range queries utilize intersection and containment operations. The distance between the query point and each point of interest is calculated by measuring the straight-line distance based on their respective coordinates. The resulting distances are then compared to the range specified by the query scope. Objects of interest that fall within this range become the query result [4]. For example, considering an area of query scope as depicted in Figure 5, the points of interest (POIs) located within that area, such as P6, P2, and P7, would be retrieved as the results of the range query (Figure 6).
In the realm of range queries, a research study explores continuous range search algorithms, highlighting the limitations of traditional methods designed for static points. The paper emphazies the inadequacy of these approaches for mobile users in motion. In response, the authors propose two methods for continuous range search, with the first one focusing on Euclidean distance [30].
Another research, conducted by Pfoser, Jensen, and Theodoridis [31] evaluated three access methods, namely, the R-Tree, Spatio-Temporal R-tree, and Trajectory-Bundle tree. The performance study involved experiments with spatial range queries, navigational queries, and combined queries.”
“2.2. Spatial queries in road network distance
According to [7], a spatial road network is fundamentally an interconnected system
of roads, illustrated as a network consisting of edges (links) and vertices (nodes). An
edge or link serves as a connection between two vertices or nodes. Each link connecting two adjacent nodes forms a segment (an unbroken link from one node to another node) without passing through any additional nodes. The distance associated with each segment is specified and referred to as the weight of that segment. The spatial road network can be utilized for conducting k-nearest neighbour (kNN) and range queries using two different approaches: restriction and expansion [7,23].”
“2.2.1. K-nearest neighbour queries in road network distance
The algorithm for kNN on spatial road networks employs two different approaches as aforementioned. The first approach, focused on restriction kNN, is the Incremental Euclidean Restriction (IER) algorithm. The IER approach involves setting an upper limit and reducing the search area. The second approach, centered on expansion kNN, is known as Incremental Network Expansion (INE). The algorithm initiates network expansion from the query point, examining entities in the order they are encountered [23].
Abeywickrama, Cheema, and Taniar [14] conduct experiment evaluating the algorithm of kNN queries on road networks include Incremental Network Expansion (INE) [23], Incremental Euclidean Restriction (IER) [23], Distance Browsing [32], Route Overlay and Association Directory (ROAD) [15, 34], and G-tree [16, 36].
In their research paper, Shahabi, Kolahdouzan, and Sharifzadeh [39], proposes a new approach for handling KNN queries in road networks, catering to both stationary and mobile query points. Their approach, named Road Network Embedding (RNE), involves converting a road network into a higher-dimensional space, allowing the use of more straightforward distance functions.
Kolahdouzan and Shahabi [37] introduce a novel approach to address the criteria for spatial query in SNDB by transforming the challenge of distance computation in a vast network into the problem of computing distances in numerous smaller networks, supplemented by additional table lookups. The central concept behind our approach, named Voronoi-based Network Nearest Neighbor (VN3), involves initially partitioning a large network into smaller and more manageable regions.
Mouratidis, Yiu, Papadias, and Mamoulis [33], the research focuses on continuous k-Nearest Neighbor (CkNN) queries in transportation networks, where data objects and queries move within a road network. While existing methods have addressed snapshot queries, continuous monitoring of CkNN queries in road networks has not been explored. The study introduces incremental monitoring and group monitoring algorithms to efficiently compute and update CkNN query results in real-time, considering fluctuations in edge weights due to factors like changing traffic conditions. The proposed algorithms aim to handle dynamic and unpredictable scenarios, such as finding the k closest taxi customers for a free taxi in terms of traveling time.”
- p. 5, line 152: better write “Taniar and Rahyau [4] categorized ...”; also briefly explain the mentioned six types.
- Thank you for your suggestion. We have revised the citation to 'Taniar and Rahyau [4]' for better clarity. Additionally, we have included a brief explanation of the six types mentioned by Taniar and Rahyau to enhance the reader's understanding. Your feedback has been valuable in improving the content, and we appreciate your attention to detail.
“2.2.2. Range queries in road network distance
In the paper by [4], It is asserted that range queries, rooted in defining a region by a specified radius or distance, can be more broadly and flexibly referred to as "region queries." These queries involve three essential components: the object of interest, the query point (representing the user’s query location), and the designated region. Taniar and Rahyau [4] categorized region queries into six types: traditional region queries, approximate region queries, constrained region queries, outer/inner fence object queries, and inverse range queries.
Traditional regions establish boundaries to extract candidate objects. Approximate region queries address situations where users may inaccurately specify the query radius due to unfamiliarity with the area. A constrained region query incorporates additional constraints, such as spatial, temporal, kNN, or others, along with the query radius. Clustered objects region queries consider objects with spatial relationships, aiming to identify clusters or groups of objects within a specified region. Inverse Region Queries involve multiple query points or objects and seek candidate objects that contain all query points within a specified range.
Turning to range queries in road networks, two different approaches are identified: restriction and extension [23]. The restriction approach for range queries called Range Queries Restriction (RER). This method conducts a range query to find objects within a specified Euclidean distance from the query point. While maintaining the Euclidean lower bound property, it reduces false hits by performing a single network expansion. The algorithm efficiently refines results using sorted lists, segment checks, and comparisons, considering boundary segments that may exceed the query threshold. On the other hand, the expansion approach for range queries called Range Network Expansion (RNE). This algorithm identifies qualifying segments within a specified network range from the query point and retrieves associated data entities. It optimizes by dividing QS into sets corresponding to R-tree entries, minimizing comparisons as it descends the tree. The algorithm ensures I/O optimality and handles scenarios where QS exceeds memory capacity through efficient methodologies and optimizations.
A study mentioned in 2.1.2 [30], which explores continuous range search in mobile navigation, concentrates on the method for moving users. , with a specific focus on moving users. The study proposes a second approach based on network distance rather than Euclidean distance, aiming for enhanced efficiency in practical applications. The outlined method involves selecting an arbitrary path from a road map, segmenting it, and utilizing Range Network Expansion (RNE) to identify entities within a specified range, considering Euclidean distance.”
“2.3. Existing research
One research study focused on exploring the relationship between transport costs
and road distance. The researchers employed a correlation factor to estimate the actual road distance by considering the Euclidean distance. They evaluated the accuracy of this approximation by calculating the average difference between the estimated and actual distances [1]. The conclusion highlights the development of a linear regression model for Euclidean distances, demonstrating its superior reliability compared to existing literature models. Two methods for adjustment are discussed: one based on mean ratios and another utilizing regression equations. The latter, while more computationally intensive, provides closer-to-reality results, but caution is urged regarding its constant, particularly for smaller distances. Notably, the study did not provide explicit information regarding the precision of the Euclidean distance. Instead, it introduced a regression model to approximate the road distance and compared it with the actual distance.
In another research study, the accuracy of the Euclidean distance was compared to the road network distance. The study investigated the performance of two query processing methods: the benchmark method, which utilized the road network distance, and the Euclidean decider, which employed the Euclidean distance [25]. The paper suggests the use of Euclidean distance as a fast and computationally less intensive alternative for estimating road network distance. The authors argue that, despite potential differences, Euclidean distance-based solutions can provide acceptable accuracy for certain queries, such as Group Nearest Neighbor (GNN) queries. Experiments involve comparing twoGNN query processing methods using road network distance as a benchmark and Euclidean Decider (using Euclidean distance). Real road network graphs from cities are utilized, creating scenarios with scattered or clustered query points and Points-Of-Interest (POIs). Parameters such as the number of query points, POI density, value of k, and road network are varied to assess performance. Euclidean Decider's accuracy is evaluated by comparing distances to benchmark results. The experiments aim to determine the effectiveness of Euclidean distance in achieving a balance between accuracy and computational efficiency for GNN queries in transportation services. The study's results [25] indicate that the Euclidean metric achieves an accuracy level exceeding 74.7% for kNN queries and exhibits even higher accuracy for general Group Nearest Neighbor (GNN) queries, ranging from 86% to 95% when considering two query points.
In a related work, Kim, Hossain, Hossain, and Chang propose Hilbert-order based star expansion cloaking algorithm (H-star) [17] is designed to optimize query processing cost by considering cloaking regions for a group of queries, reducing the number of boundary points and, consequently, the query processing cost. The proposed H-Star algorithm is further extended to include query processing algorithms for k-nearest neighbor (k-NN) queries and range queries.
Another research proposes a method for road network extraction from high-resolution synthetic aperture radar (SAR) images. The approach involves constructing road networks using smooth, cross-linked curves with determined functions, enabling mathematical descriptions for road segments. The method employs multiplicative Duda operation for line feature responses, and non-road detection techniques are introduced to reduce false positives. Binary image decomposition and polynomial curve fitting are then used to linearize road segments, and network optimization is achieved through geometric constraints [18].
In a study by Arbelaez, Mehta, O’Sullivan, Quesada, and Sasmaz [38] provides a valuable methodology for classifying exchange sites and customers into three geotypes (rural, sub-urban, and urban) based on precise customer locations, rather than relying solely on the number of connected customers. The classification is defined by household density per square kilometer, with rural areas having up to 10 households/km², sub-urban areas ranging from 11 to 500 households/km², and urban areas exceeding 501 households/km². The comprehensive distribution analysis in Ireland highlights a predominance of rural and sub-urban regions. The distribution analysis in Ireland highlights a predominance of rural and sub-urban regions, offering valuable insights for refining the classification of query points or locations in diverse scenarios.
Lastly, Boyaci, Dang, and Letchford [19] study Vehicle Routing Problems (VRPs) on road networks, which is called Steiner VRPs. Unlike traditional VRP where customers and depots are nodes in a complete graph, Steiner VRPs consider road networks, introducing challenges in terms of distance computation. The authors propose the three-phase heuristic approach: (1) create an approximation using Euclidean distances multiplied by a constant; (2) solve the approximated instance; and (3) convert the solution to the original instance. Computational experiments are conducted on Steiner versions of the Travelling Salesman Problem (TSP) and Capacitated VRP (CVRP) using real road network data from twelve cities globally. The result of this research show that the use of Euclidean distances instead of real road distances is found to yield acceptable results for Steiner Travelling Salesman Problem (TSP) and Steiner Capacitated Vehicle Routing Problem (CVRP), particularly when only a small proportion of nodes require service.”
- p. 2, line 52: explain E and V
- Thank you for pointing that out. We have now included the explanation of E and V.
“1. Introduction
The utilisation of spatial data in mobile applications has grown significantly, as it allows users to easily search for locations, navigate through unfamiliar areas, identification of transportation routes, utilisation of geomarketing techniques, and modelling environmental factors [29]. To illustrate, users can search for nearby hospitals using map applications or GPS navigation (Figure 1). To meet users’ requirements, spatial databases play a vital role in storing, retrieving, and manipulating spatial data. These databases are specifically designed to work efficiently and effectively with spatial data, allowing users to retrieve the data quickly and accurately. There are two main types of spatial queries: k-nearest neighbours (kNN) and range search. K-nearest neighbours find a specific number of nearest points of interest (POIs) based on the user’s location, while range search retrieves objects within a certain distance radius. For instance, if a user searches for the six nearest hospitals, the user’s location will be called a query point, and the map application will provide only six hospitals that are closest to the query point. On the other hand, range search differs from kNN as it is limited by distance rather than the number of spatial objects. For example, a user can search for hospitals within a 10-kilometre radius, and the application will return all the hospitals within that range.
As the usage of map applications continues to increase, spatial databases face the challenge of efficiently handling a large number of spatial queries, leading to high computational costs for retrieving spatial objects. This is because the database needs to measure the distance between the query point and all the spatial objects to identify those that meet the user’s requirements. To ensure the accuracy of retrieved spatial objects, the road network distance method is employed. This method utilizes Dijkstra’s algorithm to find the shortest path based on the actual road distance. By calculating the shortest route from the query point to the POIs, the database can accurately retrieve the desired objects. However, this algorithm introduces high computational time due to the complexity of finding the shortest distance. The time complexity of Dijkstra’s algorithm for cost functions based on vertices is approximately O(|E| + |V|log|V|) [11], where E represents the set of edges in the graph, and V represents the set of vertices. Moreover, when handling an extensive road network, the calculations required for determining the shortest paths can impose a considerable demand on both time and memory resources [19]. In scenarios such as kNN, where the total number of objects far exceeds k, computing shortest paths for all objects to identify the k-nearest neighbors becomes impractical [14]. This presents a challenge as it conflicts with the primary objective of the database, which is to enhance the retrieval process by ensuring efficiency and accuracy in data retrieval.”
- p. 8: you should explain the concept of Mesh Blocks in more detail. It remains unclear what exactly they are
- Thank you for your feedback. We acknowledge the need for a more detailed explanation of the concept of Mesh Blocks. In response to your suggestion, we have provided additional details to ensure a comprehensive understanding.
“3.1. Data source
The dataset for hospital and ambulance addresses was obtained from OpenStreetMap (OSM), a comprehensive database containing information on hospitals and ambulances across various Australian states. In contrast, a segment of the residential addresses data originated from the Geoscape Geocoded National Address File (G-NAF) [41], which adheres to the Australian Statistical Geography Standard (ASGS) [40] and encompasses addresses located within the residential Mesh Blocks (MBs) framework. Specifically, residential addresses were randomized, selecting 10 addresses from each Statistical Area Level 2 (SA2s) area, ensuring a representative sample for the study. To classify the query points, the Mesh Blocks shapefile was employed to assess their characteristics based on the density of the POIs. Mesh Blocks, as defined by the Australian Bureau of Statistics [40], are small geographical areas designed for statistical and data aggregation purposes. They offer a structured way to organize and analyze data at a fine-grained level. These Mesh Blocks served as a crucial framework for categorising the query points, enabling the evaluation of POI density in the proximity of these points. In our research, the residential address dataset was deliberately structured in accordance with Mesh Blocks, ensuring the absence of duplicate addresses. This strategic alignment serves as a precautionary measure to eliminate the potential bias that might arise from duplicate addresses within the dataset. By selecting the query point in this manner, the experimental data was able to accurately evaluate the accuracy of the Euclidean distance method in identifying the closest hospital to a given residential address.”
- On p. 9, line 275 you mention distances of 1, 5, 10, and 15 km. Why did you choose these distances?
- Thank you for your inquiry. We have included additional research in the manuscript to provide a thorough justification for the selection of distances (1 km, 5 km, 10 km, and 15 km) in section 3.3.1.1 Buffer creation as we rewrite section 3.3 Classification.
“3.3 Classification
In this section, the methodology for classifying query points based on density of POIs is explained in detail.”
“3.3.1 Criteria of classification
The classification of query points involves four distinct criteria: buffer creation, overlapping with Mesh Block, and weight calculation and utilizing the mode function in QGIS to categorize the query points based on their respective weights.”
“3.3.1.1 Buffer Creation
The first criterion is buffer creation, which involves creating spatial areas around each point of interest to classify the query points. Buffers are generated based on specified distances. In this study, we employed four different buffer distances: 1 km, 5 km, 10 km, and 15 km. The selection of these buffer distances is informed by previous research on healthcare accessibility and surveys [20-22]. These studies provide insights into acceptable distances for accessing healthcare from households and examine the relationship between distance and utilization of health services.”
“3.3.1.2 Overlapping with Mesh Block
After creating the POIs buffer with the specified distances, the criterion of overlapping with Mesh Block comes into play. Mesh Blocks containing the query points are retrieved. POIs buffer areas are overlaid with Mesh Block shapes (Figure 15 Mesh Block (red shape in the red square) POIs buffer (green and purple)). This overlapping process determines the number of weight attributes used in calculating the density of POIs. The calculation involves counting the number of buffer distances that intersect with each Mesh Block, contributing to the overall density calculation. The resulting count is stored in the column in the database (Figure 16).”
“3.3.1.3 Weight calculation and Query point classification method
The weight attributes, securely stored in the database, play a crucial role in QGIS. They are essential for calculating the density of POIs and creating a heatmap for each query point. To ensure a comprehensive representation, values from overlapping POIs buffer zones at various distances are combined. This combined value is a key parameter in the QGIS value function, necessary for creating both the heatmap and graduated visualizations. The decision to combine all POIs buffer weights is made to provide a clear and consistent representation of the overall POI density. This aggregated weight, derived from the overlap count, is also used with the data classification in QGIS, contributing to the systematic classification of query points based on their weights.
In this research, quantile classification is employed as a method for classifying query points. This approach offers several advantages, including its capacity to distribute query points uniformly across different classes, providing a balanced representation. Additionally, quantile classification is less sensitive to outliers, ensuring that extreme values do not disproportionately influence the classification results. This robustness makes it particularly suitable for scenarios where the distribution of POIs density exhibits variations, allowing for a more reliable and resilient classification process.
Considering future research directions, it may be valuable to explore alternative heatmap calculation methods, such as Kernel Density Estimation (KDE). Experimenting with different techniques could offer insights into potential variations in heatmap results. Moreover, alternative data classification methods beyond quantile classification could be explored in future investigations, enhancing the methodology's adaptability and addressing potential variations in data distribution. Discussing these possibilities in the future work section can highlight potential enhancements to the methodology and acknowledge the continuous improvement and refinement of research techniques.”
“3.3.2 Classification process
To ensure comprehensive and accurate results, 30 query points were randomly selected from the residential address dataset in each city under study, preventing clustering in a single location. This approach encourages a diverse representation of scenarios within the sample, enhancing result variability. The categorization of each query point into distinct groups based on surrounding POIs density is a pivotal step in the methodology. This categorization employs four varying buffer sizes to calculate density, assigning each point to the appropriate category.
The density calculation involves overlaying the buffer on the Mesh Block, incorporating weights derived from overlapping POIs buffer zones at different distances. The resulting values are instrumental in generating a heatmap, categorizing density into five distinct classes. Each class represents a unique density level, facilitating the observation and analysis of different scenarios. This classification process aids in evaluating the accuracy of Euclidean and road network distance methods under varying POI density conditions.
It is significant that the characteristics of POIs density vary between datasets, especially in the ambulance and hospital datasets. Despite utilizing the same query points in each dataset, the categorization may differ due to these density variations. The experiment comprises 300 samples, with 30 query points randomly selected from each city in the hospital and ambulance datasets. The cumulative count of query points in each category is as follows: Class One (73 points), Class Two (58 points), Class Three (55 points), Class Four (55 points), and Class Five (59 points). This ensures a reasonably equitable distribution, minimizing the impact of data imbalance on the results.
The balanced distribution of samples across categories enhances the reliability of experimental results, allowing for a robust evaluation of the accuracy of both Euclidean and road network distance methods. This balanced approach minimizes potential biases arising from unequal representations of different density levels, enabling a fair comparison between distance methods in diverse scenarios. The number of query points in each class is summarized in Table 2, further substantiating the experimental outcomes.”
- You never refer to Figure 14 (on p. 9) in the text. Moreover, Figure 14 and Table 1 seem to be highly redundant. I suggest omitting Fig. 14 and renumber all following figures accordingly.
- Thank you for bringing this to our attention. We acknowledge the oversight, and we have promptly removed Figure 14 as per your suggestion. The subsequent figures have been renumbered accordingly to maintain the sequential order. We appreciate your keen observation, and these changes have been implemented to enhance the clarity and coherence of the manuscript.
- On p. 10 you describe the creation of a heat map based on the Mesh Blocks. How about using KDE? What are the reason for using Mesh Blocks – despite it remains unclear what they exactly are.
- Thank you for your observation. We acknowledge the question regarding the use of Mesh Blocks and the suggestion of using Kernel Density Estimation (KDE) for creating a heatmap. We appreciate your feedback, and we want to clarify that the methodology we employed to plot the heatmap utilized QGIS, which does not inherently support KDE. We will address this limitation and consider the use of KDE in our future work. Additionally, we will provide more clarity on Mesh Blocks in the limitation and future work in ‘Discussion’ and ‘Conclusion’ sections.
- I would expect a deeper reflection in the discussion section. What does this all
- Thank you for your feedback. We have acknowledged the lack of depth analysis and discussion. Hence, we have improved in analysis and provide practical implications of the findings in the ‘Discussion’ section.
- mean? What follows from your results. Can you make any recommendations? In the current state, the paper kind of remains unfinished and lacks a clear contribution.
- Thank you for your insightful comment. We recognize the need for a more explicit connection between our results and their implications. In response to your feedback, we have revisited the discussion section to draw clearer conclusions from our results and provide meaningful recommendations.
“5. Discussion
The primary objective of this research was to evaluate the accuracy of the Euclidean distance method in comparison to the road network distance method, the Open Source Routing Machine (OSRM) was utilized to extract the road network distances, which served as the benchmark for comparison with the Euclidean distance method. OSRM provided a reliable and efficient means of calculating the actual distances based on the road network topology. By using OSRM as the benchmark, we were able to assess the accuracy of the Euclidean distance method in capturing the true distances along the road network. Furthermore, the study aimed to comprehensively examine the unique characteristics associated with each classification of query points and rigorously evaluate the accuracy of each classification. Our analysis revealed the accuracy rate for each query method on the Euclidean distance compared with the road network distance, which was set as the benchmark. In kNN queries, we observed a correlation between the accuracy and the number of k. Specifically, we found that as the number of k increased, the accuracy for each class of query points consistently decreased.This observed could be attributed to the intricate road structure within the city. As the value of k rises, the algorithm considers a higher number of nearby points, potentially leading to a greater diversity in road structures and terrain. This diversity might introduce variations in the distances calculated, impacting the accuracy of the Euclidean distance method. Additionally, the road network's complexity, including factors like one-way streets, intersections, and irregularities, could contribute to this phenomenon. This finding highlights the impact of the chosen value of k on the accuracy of the Euclidean distance method. Regarding the range query, we identified a consistent pattern across different classes, with each class exhibiting a similar trend except for the turning point in the range value. The turning point, which represents the threshold where the accuracy starts to increase, appears to be influenced by the density of POIs. Class Five showed the earliest increase in accuracy, followed by class Four, suggesting a relationship between class and the point at which accuracy begins to improve.
However, the observed increase in accuracy in this context may suggest that the distance range adequately covers all surrounding Points of Interest (POIs). As previously mentioned in the kNN case, where road structure influenced the decrease in accuracy, here we see a correlated phenomenon. The high-density Class experiences a decrease in accuracy when the range distance is less, indicating a potential interplay between road structure, density, and range distance that impacts accuracy trends. Another insight gained from this research is that a high number of POIs does not necessarily guarantee high accuracy. Despite the hospital dataset having fewer POIs compared to the ambulance dataset, the accuracy of the hospital dataset was found to be higher. This observation suggests that when there are limited choices (few POIs), the retrieved results using the Euclidean distance method may closely align with those obtained from the road network. Nevertheless, as the number of available choices increases, there is a likelihood of encountering slight differences in distances, potentially influencing the final results and impacting overall accuracy. This significant finding has the potential to substantially contribute to future research endeavors focused on analyzing the conditions under which the Euclidean distance can be employed as a viable trade-off between accuracy and computational time, particularly considering the classification of query points based on POIs density. By highlighting the circumstances in which this alternative metric performs acceptably across different density classes, our study provides valuable insights for practical implementations. This understanding becomes particularly crucial for scenarios where the computational efficiency gained from using Euclidean distance outweighs the marginal decrease in accuracy, offering a practical solution for certain types of queries in real-world applications.
Previous studies have extensively investigated the accuracy of the Euclidean distance method and have used the road network distance method as a benchmark. Notably, Hua, Xie, and Tanin (2018) employ Euclidean distance as a computationally efficient alternative for estimating road network distance, specifically in the context of Group Nearest Neighbor (GNN) queries. The experiments involve comparing the Euclidean Decider, which utilizes Euclidean distance, with road network distance as a benchmark. Real road network graphs from various cities are used to create scenarios with different query point and POI distributions. The study varies parameters such as the number of query points, POI density, value of k, and road network structure. The results indicate that the Euclidean metric achieves high accuracy levels, exceeding 74.7% for kNN queries and ranging from 86% to 95% for general GNN queries, showcasing its effectiveness in achieving a balance between accuracy and computational efficiency in transportation services.
In comparison, our studies demonstrated high accuracy at certain values of k when using kNN queries and achieved an accuracy rate above 50% for range queries. The results from their approach for kNN queries showed significantly higher accuracy. This difference may arise due to the influence of various parameters or variables affecting accuracy. The substantial contrast in outcomes could be attributed to specific conditions, configurations, or factors inherent in their approach compared to ours. Further investigation into the specific variables or parameters that contribute to this marked variation in accuracy would be essential for a comprehensive understanding of the results.
In our study, we categorized the query points by utilizing the attribute weight derived from counting the number of POIs buffer that overlaid with the Mesh Block containing the query points. This approach enabled us to use the weight attribute to calculate the density using a heatmap. However, it is important to acknowledge several limitations that may impact the accuracy of the result. Firstly, the criteria for query point classification, including the selection of the weight attribute, the method employed for heatmap calculation, and geo-types, can introduce variability into the results. Additionally, The road structure for each city potentially affects the result of the POIs for the road network distance method which might impact the accuracy of the Euclidean distance Furthermore, we recognize that there may be an imbalance in the number of POIs across different cities. This could potentially introduce bias in the analysis, as certain cities may have a higher concentration of POIs compared to others. This imbalance could influence the results, particularly when comparing the accuracy of the Euclidean distance method across different cities.
By acknowledging these limitations, we demonstrate a critical awareness of the potential factors that could influence the accuracy and reliability of our study’s results.”
“6. Conclusion
In conclusion, our study aimed to evaluate the accuracy of the Euclidean distance method in comparison to the road network distance method using kNN queries and range queries, considering the classification of query points. The investigation brought several significant insights.
Our findings unveiled several critical insights. Specifically, in kNN queries, peak accuracy was achieved when k was set to 1, exceeding 85% across all classes. However, with an increase in k, a consistent decrease in accuracy was observed, highlighting the sensitivity of results to the choice of k. Turning to range queries, a uniform accuracy pattern emerged across classes, with the turning point influenced by the varying range values. Notably, a higher density of POIs correlated with an earlier increase in accuracy.
Furthermore, the comparative analysis between ambulance and hospital datasets questioned the assumption that a greater number of POIs ensures superior accuracy. Despite containing fewer POIs, the hospital dataset demonstrated higher accuracy, emphasizing the complex interplay between dataset characteristics and method performance. Our study provides a substantial contribution by illuminating scenarios in which the Euclidean distance method serves as a feasible compromise between accuracy and computational efficiency, particularly in the classification of query points based on POI density.
The discussion provided a detailed understanding of the relationship between accuracy and the number of k in kNN queries. As k increased, diversity in road structures and terrain introduced variations, impacting accuracy. Similarly, in range queries, the observed increase in accuracy suggested that the distance range adequately covered surrounding POIs. However, the study cautioned about potential interactions between road structure, POIs density, and range distance influencing accuracy trends.
While our approach adds valuable insights to the existing body of research, we acknowledge certain limitations. The criteria for query point classification, the selection of the weight attribute, heatmap calculation method, and consideration of different geo-types introduce variability into our results. The impact of road structure on the accuracy of POIs in the road network distance method and potential bias due to an imbalance in the number of POIs across cities are factors that could influence the accuracy comparison of the Euclidean distance method.
In future work, accounting for road structure in query point classification and addressing POI imbalances across cities will refine our understanding of the Euclidean distance method's performance. Additionally, exploring the impact of criteria for classification, the number of samples, the road network algorithm for extracting the shortest route, the weight attribute for heatmap calculation, and different geo-types (dense urban areas, sparse rural areas) will provide a more comprehensive assessment. By doing so, we aim to enhance the applicability of our findings to real-world scenarios while mitigating potential biases and improving the robustness of our methodology.”
- The classifications in Figures 18 – 21 have overlapping classes (e.g., 0 – 6, 6 – 9, ...). Please adjust the class breaks.
- We have made an adjustment regarding your comments. Thank you for pointing this out.
- For Figures 25 and 27 you should increase the contrast of the colour hues, so it becomes easier to distinguish the different categories (hospitals, ambulances , all).
- Thank you for your observation. We acknowledge that the color hues may not have been clear enough to distinguish between the different categories in Figures 25 and 27. In response, we have provided distinctions in the line graph to ensure that readers can easily discern the differences in each category. We appreciate your feedback, and these changes are aimed at enhancing the visual clarity of our figures.
- Some minor language issues are:
- 7, line 189/190: The verb seems to be missing in this sentence.
- 7, line 213: in 13 -> in Figure 13
- 14., line 345: ... distance between query points and POIs calculate on the ... -> ... distance between query points and POIs were calculated on the
- 15, line 399: closely aligns with the Euclidean distance method. -> This sentence
- is incomplete; please rephrase.
- 16, line 452: this step -> these steps
- 16, line 469: figure 23 -> Figure 23
- 17, line 489: in this sentence the verb seems to be missing
- 17, line 492 / p. 18, line 493: Hence, it is expecting the accuracy results for each
- value of k different between the two datasets. -> Hence, it is expected that the
- accuracy results for each value of k differ between the two datasets.
- 18, line 501: ... method result compare with the benchmark ... -> ... method result compared with the benchmark ...
- Thank you for highlighting the language issues. We appreciate your feedback, and we have made the necessary changes to address the minor language issues in the manuscript. Below are the examples of change in language issues highlight in yellow.
“3. Methodology
To evaluate the accuracy of the Euclidean distance method, this research employed a quantitative approach. Initially, a dataset was compiled, consisting of hospital and ambulance addresses as points of interest (POIs), along with residential addresses for query points. Following data collection, the dataset was divided into hospital and ambulance addresses. Subsequently, a data exploration process was undertaken, revealing that only five cities had a sufficient number of points of interest to conduct the experiment. Therefore, 30 query points were randomly selected from each of these five cities to perform the research. This careful process guaranteed a thorough evaluation of how accurate the Euclidean distance method is in the selected urban areas we studied. The query points were further categorized based on quantitative criteria, primarily the density of Points of Interest (POIs), determined using attribute weights for precise calculations. This classification facilitated a systematic analysis of the data. In the subsequent step, both Euclidean distance and the Road Network Distance method were employed for POI retrieval, utilizing kNN and range search techniques. The accuracy of the Euclidean distance method was then assessed by comparing its outcomes with those derived from the road network distance method, which served as a reliable quantitative benchmark. Notably, the road network distance method provided accurate distance calculations based on real road networks. In this study, a systematic approach was followed as depicted in Figure 13, where the methodology flow chart illustrates the step-by-step process. By conducting thorough quantitative analyses and carefully comparing the results, this research aimed to evaluate the accuracy of the Euclidean distance method in retrieving spatial data, ensuring a comprehensive and reliable assessment.”
“3.4.1 kNN search in Euclidean distance
The kNN queries method searched for the nearest POIs to the query point based on the number of points specified by the user, which was set limited to the first 20 data points (k = [1,20]). These points are sorted based on their distance from the query point. The distance between query points and POIs were calculated on the database following this statement:”
“3.5.1 kNN search in Road network distance
The process of obtaining results in kNN search using the road network distance method closely aligns with the Euclidean distance method. However, a key difference lies in the approach: the Euclidean distance method calculates the direct distance between a query point and points of interest using a spatial database. On the other hand, in the road network distance method, distances are acquired by utilizing the Open Source Routing Machine (OSRM) API through Python code. This API extracts the shortest route, and the resulting distances are then stored in a specified database table.”
“3.6.2 Range queries
For the range query, this method retrieved all POIs located within the specified distance range, which was set to 5, 10, 15, 20, 25, 30, 35, …, and 150 km, and the results were sorted based on their distance from the query point. As the number of retrieved POIs was not limited in the range search method, the percentage of accuracy was calculated by following these steps:
The accuracy results for each category were plotted and are presented in Figure 22. It's interesting to observe that as the value of k increases, the accuracy of the classification decreases across all categories. Even for query points with the highest density of POIs, the accuracy still decreases. The results indicate that the accuracy remains high for classes one and three, with rates above 85\%, as well as for the remaining classes, which have rates exceeding 90\%. This suggests that the Euclidean distance method and road network distance method consistently provide accurate results, particularly for the first category of points of interest.
Analysing Figure 22 and 23, we observe the results obtained from queries in both types of POIs. It is crucial to consider the differences between 2 different types of dataset since the number of POIs are significantly different between them. The hospital dataset contains less data points than the ambulance dataset. Hence, it is expected that the accuracy results for each value of k differ between the two datasets. By taking into account these differences, we can develop a more comprehensive insight into how dataset attributes influence the accuracy of classification outcomes. This examination enables us to uncover the nuances that distinguish the hospital dataset from the ambulance dataset, offering valuable insights for future enhancements and similar analyses.
Figure 24 shows line graphs that plot the accuracy separately for the hospital dataset, ambulance dataset, and a combined dataset. Each class is represented in its own subplot. The purpose of the graph is to observe the accuracy of each dataset of the Euclidean distance method result compared with the benchmark as the result of the road network distance method.”
Reviewer 2 Report
Comments and Suggestions for Authors
In this paper the authors study the trade-off between the time complexity and the accuracy in spatial query processing. Furthermore, it addresses certain challenges associated with retrieving points of interest while using the actual road distance and the benefits of the traditional approach based on the Euclidean distance.
As expected the evaluation suggests that the accuracy decreases as k increases.
In general, the paper is well written and easy to follow. However, the authors need to extend the related work section with state-of-the-art approaches to tackle the trade-off between the time complexity and the accuracy of calculating the distance with road networks.
For instance, “Alejandro Arbelaez, Deepak Mehta, Barry O'Sullivan, Luis Quesada, Ata Sasmaz: Generation of a reference network for Ireland and its contribution to the design of an optical network architecture. ICTON 2017: 1-4” proposed a methodology to substantially reduce the size of the reference map (e.g., from open street maps) and compute the distance between two points.
Alternatively, “Xiao, F.; Tong, L.; Luo, S. A Method for Road Network Extraction from High-Resolution SAR Imagery Using Direction Grouping and Curve Fitting. Remote Sens. 2019,” also proposes a method for extracting reference road networks from images.
Also “Kim, Yong‐Ki, et al. "Hilbert‐order based spatial cloaking algorithm in road network." Concurrency and Computation: Practice and Experience 25.1 (2013): 143-158.” Uses the hilbert curve for road network design.
In general, the authors need to substantially increase the related work section by taking into account the literature in the area such as the three above mentioned papers and many more, e.g.,
Kolahdouzan M, Shahabi C. Voronoi-based k nearest neighbor search for spatial network databases. Proceeding of 30th International Conference Very Large Data Bases (VLDB), Vol. 30, 2004; 840–851.
Mouratidis K, Yiu M, Papadias D, Mamoulis N. Continuous nearest neighbor monitoring in road networks. Proceeding of 32th International Conference Very Large Data Bases (VLDB) 2006: 43–54.
Furthermore, it would be important to use different geotypes in the evaluation approach (dense urban areas, spare rural areas, etc.) to fully assess the accuracy of the current methodology.
Comments on the Quality of English Language
No major comments
Author Response
Dear Reviewer 2:
Thank you for your feedback on ‘Navigating the Maps: Euclidean vs. Road Network Distances in Spatial Queries.’ We have revised the manuscript to incorporate most of the suggestions provided by the Reviewer 2, and these changes are highlighted within the text. Your guidance has been invaluable, and we appreciate the opportunity to enhance the quality of our work based on your insights.
Best regard,
Pornrawee Tatit, Kiki Maulana Adhinugraha, David Taniar
Comment from Reviewer 2:
- The authors need to extend the related work section with state-of-the-art approaches to tackle the trade-off between the time complexity and the accuracy of calculating the distance with road networks.
- Thank you for your valuable feedback and constructive suggestions regarding the related work section. We fully recognize the importance of extending the discussion to encompass state-of-the-art approaches addressing the trade-off between time complexity and accuracy in distance calculations with road networks. In our revision, we have diligently expanded the 'Existing Research' section to incorporate additional studies that specifically relate to our research focus on the trade-off between time complexity and accuracy in distance calculations with road networks. These recently added studies contribute to a more refined understanding of the current state of the field and enhance the context of our research within the broader literature. Additionally, the yellow highlights indicate additional content or the changes from the previous manuscript.
- In general, the authors need to substantially increase the related work section by taking into account the literature in the area such as the three above mentioned papers and many more, e.g.,
- Thank you for your thorough review and insightful feedback on the related work section. We appreciate your suggestion to substantially increase the coverage of relevant literature, including the three papers you mentioned and others within the research domain. In response to your guidance, we significantly expand the ‘Related work’ section to encompass a more comprehensive review of the existing literature, giving due consideration to the three papers you highlighted and incorporating additional relevant studies.
“2. Related work
2.1. Spatial queries in Euclidean distance
Spatial data objects represent a type of data that encompasses multiple dimensions. Over the past few years, there has been a growing utilization of this data type in various applications, including geographic information systems (GIS), computer-aided design, and computer vision [8]. The commonly employed methods for searching spatial objects include k-nearest neighbours (kNN) and range queries techniques.”
“2.1.1. K-nearest neighbour queries in Euclidean distance
The effective execution of Nearest Neighbor (NN) queries, which holds special significance in the field of Geographic Information Systems (GIS) [28], is particularly important when considering the commonly used query type known as k-Nearest Neighbour (kNN). The kNN query involves finding the potential neighbours closest to a given query point [7,28]. In the process of retrieving Points of Interest (POIs) using k-Nearest Neighbours, the distance between the query point and the POIs is calculated by measuring the straight-line distance from the query point’s coordinates to the coordinates of each POI. The resulting POIs are then ordered based on their distance from the query point, enabling the selection of only the top k-ranked POIs from the results. To illustrate, Figure 2 depicts a map diagram with weighted lines representing the actual distance of the road. By employing the Euclidean distance method, the distances are measured as shown in Figure 3. As a result, the retrieved POIs from the Euclidean distance method would be P6, P2, and P7, as indicated in Figure 4.
Fukunaga and Narendra [9] proposed an effective method for computing k-nearest neighbors. Their approach involves a hierarchical decomposition of design samples into disjoint subsets. To achieve this, they applied the branch and bound algorithm, a well-regarded tree-search technique [10, 12, 13]. This algorithm is known for its effectiveness in efficiently searching through the resultant groups. Expanding the scope of k-nearest neighbor (kNN) query processing, Taniar and Rahayu [7] discuss the extension of k-nearest neighbor (kNN) query processing to include obstacles in Euclidean space, known as Obstacle Nearest Neighbors (ONN). ONN involves constructing a visibility graph, representing possible paths around obstacles, and using a disk-based approach with a Restriction method to efficiently process kNN queries while considering obstacle distances. The method iteratively refines search spaces based on obstacle distances to optimize query processing.
Additionally, various types of Nearest Neighbor queries have been studied, including aggregate nearest neighbor (ANN) [3], reverse nearest neighbor (RNN) [27], and group nearest neighbor (GNN) [26]. However, for our specific study, we will focus on the k-nearest neighbor (kNN) queries.”
“2.1.2. Range queries in Euclidean distance
Range queries are an essential query method in database systems [2]. It builds upon the concept of a point query, but in range queries, the area around the query point is expanded, forming an area of query scope. Range queries involve three primary components: points of interest, the query point, and the query scope. To retrieve the relevant objects, range queries utilize intersection and containment operations. The distance between the query point and each point of interest is calculated by measuring the straight-line distance based on their respective coordinates. The resulting distances are then compared to the range specified by the query scope. Objects of interest that fall within this range become the query result [4]. For example, considering an area of query scope as depicted in Figure 5, the points of interest (POIs) located within that area, such as P6, P2, and P7, would be retrieved as the results of the range query (Figure 6).
In the realm of range queries, a research study explores continuous range search algorithms, highlighting the limitations of traditional methods designed for static points. The paper emphazies the inadequacy of these approaches for mobile users in motion. In response, the authors propose two methods for continuous range search, with the first one focusing on Euclidean distance [30].
Another research, conducted by Pfoser, Jensen, and Theodoridis [31] evaluated three access methods, namely, the R-Tree, Spatio-Temporal R-tree, and Trajectory-Bundle tree. The performance study involved experiments with spatial range queries, navigational queries, and combined queries.”
“2.2. Spatial queries in road network distance
According to [7], a spatial road network is fundamentally an interconnected system
of roads, illustrated as a network consisting of edges (links) and vertices (nodes). An
edge or link serves as a connection between two vertices or nodes. Each link connecting two adjacent nodes forms a segment (an unbroken link from one node to another node) without passing through any additional nodes. The distance associated with each segment is specified and referred to as the weight of that segment. The spatial road network can be utilized for conducting k-nearest neighbour (kNN) and range queries using two different approaches: restriction and expansion [7,23].”
“2.2.1. K-nearest neighbour queries in road network distance
The algorithm for kNN on spatial road networks employs two different approaches as aforementioned. The first approach, focused on restriction kNN, is the Incremental Euclidean Restriction (IER) algorithm. The IER approach involves setting an upper limit and reducing the search area. The second approach, centered on expansion kNN, is known as Incremental Network Expansion (INE). The algorithm initiates network expansion from the query point, examining entities in the order they are encountered [23].
Abeywickrama, Cheema, and Taniar [14] conduct experiment evaluating the algorithm of kNN queries on road networks include Incremental Network Expansion (INE) [23], Incremental Euclidean Restriction (IER) [23], Distance Browsing [32], Route Overlay and Association Directory (ROAD) [15, 34], and G-tree [16, 36].
In their research paper, Shahabi, Kolahdouzan, and Sharifzadeh [39], proposes a new approach for handling KNN queries in road networks, catering to both stationary and mobile query points. Their approach, named Road Network Embedding (RNE), involves converting a road network into a higher-dimensional space, allowing the use of more straightforward distance functions.
Kolahdouzan and Shahabi [37] introduce a novel approach to address the criteria for spatial query in SNDB by transforming the challenge of distance computation in a vast network into the problem of computing distances in numerous smaller networks, supplemented by additional table lookups. The central concept behind our approach, named Voronoi-based Network Nearest Neighbor (VN3), involves initially partitioning a large network into smaller and more manageable regions.
Mouratidis, Yiu, Papadias, and Mamoulis [33], the research focuses on continuous k-Nearest Neighbor (CkNN) queries in transportation networks, where data objects and queries move within a road network. While existing methods have addressed snapshot queries, continuous monitoring of CkNN queries in road networks has not been explored. The study introduces incremental monitoring and group monitoring algorithms to efficiently compute and update CkNN query results in real-time, considering fluctuations in edge weights due to factors like changing traffic conditions. The proposed algorithms aim to handle dynamic and unpredictable scenarios, such as finding the k closest taxi customers for a free taxi in terms of traveling time. “
“2.2.2. Range queries in road network distance
In the paper by [4], It is asserted that range queries, rooted in defining a region by a specified radius or distance, can be more broadly and flexibly referred to as "region queries." These queries involve three essential components: the object of interest, the query point (representing the user’s query location), and the designated region. Taniar and Rahyau [4] categorized region queries into six types: traditional region queries, approximate region queries, constrained region queries, outer/inner fence object queries, and inverse range queries.
Traditional regions establish boundaries to extract candidate objects. Approximate region queries address situations where users may inaccurately specify the query radius due to unfamiliarity with the area. A constrained region query incorporates additional constraints, such as spatial, temporal, kNN, or others, along with the query radius. Clustered objects region queries consider objects with spatial relationships, aiming to identify clusters or groups of objects within a specified region. Inverse Region Queries involve multiple query points or objects and seek candidate objects that contain all query points within a specified range.
Turning to range queries in road networks, two different approaches are identified: restriction and extension [23]. The restriction approach for range queries called Range Queries Restriction (RER). This method conducts a range query to find objects within a specified Euclidean distance from the query point. While maintaining the Euclidean lower bound property, it reduces false hits by performing a single network expansion. The algorithm efficiently refines results using sorted lists, segment checks, and comparisons, considering boundary segments that may exceed the query threshold. On the other hand, the expansion approach for range queries called Range Network Expansion (RNE). This algorithm identifies qualifying segments within a specified network range from the query point and retrieves associated data entities. It optimizes by dividing QS into sets corresponding to R-tree entries, minimizing comparisons as it descends the tree. The algorithm ensures I/O optimality and handles scenarios where QS exceeds memory capacity through efficient methodologies and optimizations.
A study mentioned in 2.1.2 [30], which explores continuous range search in mobile navigation, concentrates on the method for moving users. , with a specific focus on moving users. The study proposes a second approach based on network distance rather than Euclidean distance, aiming for enhanced efficiency in practical applications. The outlined method involves selecting an arbitrary path from a road map, segmenting it, and utilizing Range Network Expansion (RNE) to identify entities within a specified range, considering Euclidean distance.”
“2.3. Existing research
One research study focused on exploring the relationship between transport costs
and road distance. The researchers employed a correlation factor to estimate the actual road distance by considering the Euclidean distance. They evaluated the accuracy of this approximation by calculating the average difference between the estimated and actual distances [1]. The conclusion highlights the development of a linear regression model for Euclidean distances, demonstrating its superior reliability compared to existing literature models. Two methods for adjustment are discussed: one based on mean ratios and another utilizing regression equations. The latter, while more computationally intensive, provides closer-to-reality results, but caution is urged regarding its constant, particularly for smaller distances. Notably, the study did not provide explicit information regarding the precision of the Euclidean distance. Instead, it introduced a regression model to approximate the road distance and compared it with the actual distance.
In another research study, the accuracy of the Euclidean distance was compared to the road network distance. The study investigated the performance of two query processing methods: the benchmark method, which utilized the road network distance, and the Euclidean decider, which employed the Euclidean distance [25]. The paper suggests the use of Euclidean distance as a fast and computationally less intensive alternative for estimating road network distance. The authors argue that, despite potential differences, Euclidean distance-based solutions can provide acceptable accuracy for certain queries, such as Group Nearest Neighbor (GNN) queries. Experiments involve comparing twoGNN query processing methods using road network distance as a benchmark and Euclidean Decider (using Euclidean distance). Real road network graphs from cities are utilized, creating scenarios with scattered or clustered query points and Points-Of-Interest (POIs). Parameters such as the number of query points, POI density, value of k, and road network are varied to assess performance. Euclidean Decider's accuracy is evaluated by comparing distances to benchmark results. The experiments aim to determine the effectiveness of Euclidean distance in achieving a balance between accuracy and computational efficiency for GNN queries in transportation services. The study's results [25] indicate that the Euclidean metric achieves an accuracy level exceeding 74.7% for kNN queries and exhibits even higher accuracy for general Group Nearest Neighbor (GNN) queries, ranging from 86% to 95% when considering two query points.
In a related work, Kim, Hossain, Hossain, and Chang propose Hilbert-order based star expansion cloaking algorithm (H-star) [17] is designed to optimize query processing cost by considering cloaking regions for a group of queries, reducing the number of boundary points and, consequently, the query processing cost. The proposed H-Star algorithm is further extended to include query processing algorithms for k-nearest neighbor (k-NN) queries and range queries.
Another research proposes a method for road network extraction from high-resolution synthetic aperture radar (SAR) images. The approach involves constructing road networks using smooth, cross-linked curves with determined functions, enabling mathematical descriptions for road segments. The method employs multiplicative Duda operation for line feature responses, and non-road detection techniques are introduced to reduce false positives. Binary image decomposition and polynomial curve fitting are then used to linearize road segments, and network optimization is achieved through geometric constraints [18].
In a study by Arbelaez, Mehta, O’Sullivan, Quesada, and Sasmaz [38] provides a valuable methodology for classifying exchange sites and customers into three geotypes (rural, sub-urban, and urban) based on precise customer locations, rather than relying solely on the number of connected customers. The classification is defined by household density per square kilometer, with rural areas having up to 10 households/km², sub-urban areas ranging from 11 to 500 households/km², and urban areas exceeding 501 households/km². The comprehensive distribution analysis in Ireland highlights a predominance of rural and sub-urban regions. The distribution analysis in Ireland highlights a predominance of rural and sub-urban regions, offering valuable insights for refining the classification of query points or locations in diverse scenarios.
Lastly, Boyaci, Dang, and Letchford [19] study Vehicle Routing Problems (VRPs) on road networks, which is called Steiner VRPs. Unlike traditional VRP where customers and depots are nodes in a complete graph, Steiner VRPs consider road networks, introducing challenges in terms of distance computation. The authors propose the three-phase heuristic approach: (1) create an approximation using Euclidean distances multiplied by a constant; (2) solve the approximated instance; and (3) convert the solution to the original instance. Computational experiments are conducted on Steiner versions of the Travelling Salesman Problem (TSP) and Capacitated VRP (CVRP) using real road network data from twelve cities globally. The result of this research show that the use of Euclidean distances instead of real road distances is found to yield acceptable results for Steiner Travelling Salesman Problem (TSP) and Steiner Capacitated Vehicle Routing Problem (CVRP), particularly when only a small proportion of nodes require service.”
- It would be important to use different geotypes in the evaluation approach (dense urban areas, spare rural areas, etc.) to fully assess the accuracy of the current methodology.
- Thank you for your insightful comment on the evaluation approach. We appreciate your suggestion to incorporate different geotypes, such as dense urban areas and sparse rural areas, in our assessment methodology. Recognizing the importance of this consideration, we plan to integrate a diverse set of geotypes to ensure a more comprehensive evaluation of the accuracy of our methodology. Hence, we have addressed this limitation and outlined the plan for future work in the 'Discussion' and 'Conclusion' sections.
“5. Discussion
The primary objective of this research was to evaluate the accuracy of the Euclidean distance method in comparison to the road network distance method, the Open Source Routing Machine (OSRM) was utilized to extract the road network distances, which served as the benchmark for comparison with the Euclidean distance method. OSRM provided a reliable and efficient means of calculating the actual distances based on the road network topology. By using OSRM as the benchmark, we were able to assess the accuracy of the Euclidean distance method in capturing the true distances along the road network. Furthermore, the study aimed to comprehensively examine the unique characteristics associated with each classification of query points and rigorously evaluate the accuracy of each classification. Our analysis revealed the accuracy rate for each query method on the Euclidean distance compared with the road network distance, which was set as the benchmark. In kNN queries, we observed a correlation between the accuracy and the number of k. Specifically, we found that as the number of k increased, the accuracy for each class of query points consistently decreased.This observed could be attributed to the intricate road structure within the city. As the value of k rises, the algorithm considers a higher number of nearby points, potentially leading to a greater diversity in road structures and terrain. This diversity might introduce variations in the distances calculated, impacting the accuracy of the Euclidean distance method. Additionally, the road network's complexity, including factors like one-way streets, intersections, and irregularities, could contribute to this phenomenon. This finding highlights the impact of the chosen value of k on the accuracy of the Euclidean distance method. Regarding the range query, we identified a consistent pattern across different classes, with each class exhibiting a similar trend except for the turning point in the range value. The turning point, which represents the threshold where the accuracy starts to increase, appears to be influenced by the density of POIs. Class Five showed the earliest increase in accuracy, followed by class Four, suggesting a relationship between class and the point at which accuracy begins to improve.
However, the observed increase in accuracy in this context may suggest that the distance range adequately covers all surrounding Points of Interest (POIs). As previously mentioned in the kNN case, where road structure influenced the decrease in accuracy, here we see a correlated phenomenon. The high-density Class experiences a decrease in accuracy when the range distance is less, indicating a potential interplay between road structure, density, and range distance that impacts accuracy trends. Another insight gained from this research is that a high number of POIs does not necessarily guarantee high accuracy. Despite the hospital dataset having fewer POIs compared to the ambulance dataset, the accuracy of the hospital dataset was found to be higher. This observation suggests that when there are limited choices (few POIs), the retrieved results using the Euclidean distance method may closely align with those obtained from the road network. Nevertheless, as the number of available choices increases, there is a likelihood of encountering slight differences in distances, potentially influencing the final results and impacting overall accuracy. This significant finding has the potential to substantially contribute to future research endeavors focused on analyzing the conditions under which the Euclidean distance can be employed as a viable trade-off between accuracy and computational time, particularly considering the classification of query points based on POIs density. By highlighting the circumstances in which this alternative metric performs acceptably across different density classes, our study provides valuable insights for practical implementations. This understanding becomes particularly crucial for scenarios where the computational efficiency gained from using Euclidean distance outweighs the marginal decrease in accuracy, offering a practical solution for certain types of queries in real-world applications.
Previous studies have extensively investigated the accuracy of the Euclidean distance method and have used the road network distance method as a benchmark. Notably, Hua, Xie, and Tanin (2018) employ Euclidean distance as a computationally efficient alternative for estimating road network distance, specifically in the context of Group Nearest Neighbor (GNN) queries. The experiments involve comparing the Euclidean Decider, which utilizes Euclidean distance, with road network distance as a benchmark. Real road network graphs from various cities are used to create scenarios with different query point and POI distributions. The study varies parameters such as the number of query points, POI density, value of k, and road network structure. The results indicate that the Euclidean metric achieves high accuracy levels, exceeding 74.7% for kNN queries and ranging from 86% to 95% for general GNN queries, showcasing its effectiveness in achieving a balance between accuracy and computational efficiency in transportation services.
In comparison, our studies demonstrated high accuracy at certain values of k when using kNN queries and achieved an accuracy rate above 50% for range queries. The results from their approach for kNN queries showed significantly higher accuracy. This difference may arise due to the influence of various parameters or variables affecting accuracy. The substantial contrast in outcomes could be attributed to specific conditions, configurations, or factors inherent in their approach compared to ours. Further investigation into the specific variables or parameters that contribute to this marked variation in accuracy would be essential for a comprehensive understanding of the results.
In our study, we categorized the query points by utilizing the attribute weight derived from counting the number of POIs buffer that overlaid with the Mesh Block containing the query points. This approach enabled us to use the weight attribute to calculate the density using a heatmap. However, it is important to acknowledge several limitations that may impact the accuracy of the result. Firstly, the criteria for query point classification, including the selection of the weight attribute, the method employed for heatmap calculation, and geo-types, can introduce variability into the results. Additionally, The road structure for each city potentially affects the result of the POIs for the road network distance method which might impact the accuracy of the Euclidean distance Furthermore, we recognize that there may be an imbalance in the number of POIs across different cities. This could potentially introduce bias in the analysis, as certain cities may have a higher concentration of POIs compared to others. This imbalance could influence the results, particularly when comparing the accuracy of the Euclidean distance method across different cities.
By acknowledging these limitations, we demonstrate a critical awareness of the potential factors that could influence the accuracy and reliability of our study’s results.”
“6. Conclusion
In conclusion, our study aimed to evaluate the accuracy of the Euclidean distance method in comparison to the road network distance method using kNN queries and range queries, considering the classification of query points. The investigation brought several significant insights.
Our findings unveiled several critical insights. Specifically, in kNN queries, peak accuracy was achieved when k was set to 1, exceeding 85% across all classes. However, with an increase in k, a consistent decrease in accuracy was observed, highlighting the sensitivity of results to the choice of k. Turning to range queries, a uniform accuracy pattern emerged across classes, with the turning point influenced by the varying range values. Notably, a higher density of POIs correlated with an earlier increase in accuracy.
Furthermore, the comparative analysis between ambulance and hospital datasets questioned the assumption that a greater number of POIs ensures superior accuracy. Despite containing fewer POIs, the hospital dataset demonstrated higher accuracy, emphasizing the complex interplay between dataset characteristics and method performance. Our study provides a substantial contribution by illuminating scenarios in which the Euclidean distance method serves as a feasible compromise between accuracy and computational efficiency, particularly in the classification of query points based on POI density.
The discussion provided a detailed understanding of the relationship between accuracy and the number of k in kNN queries. As k increased, diversity in road structures and terrain introduced variations, impacting accuracy. Similarly, in range queries, the observed increase in accuracy suggested that the distance range adequately covered surrounding POIs. However, the study cautioned about potential interactions between road structure, POIs density, and range distance influencing accuracy trends.
While our approach adds valuable insights to the existing body of research, we acknowledge certain limitations. The criteria for query point classification, the selection of the weight attribute, heatmap calculation method, and consideration of different geo-types introduce variability into our results. The impact of road structure on the accuracy of POIs in the road network distance method and potential bias due to an imbalance in the number of POIs across cities are factors that could influence the accuracy comparison of the Euclidean distance method.
In future work, accounting for road structure in query point classification and addressing POI imbalances across cities will refine our understanding of the Euclidean distance method's performance. Additionally, exploring the impact of criteria for classification, the number of samples, the road network algorithm for extracting the shortest route, the weight attribute for heatmap calculation, and different geo-types (dense urban areas, sparse rural areas) will provide a more comprehensive assessment. By doing so, we aim to enhance the applicability of our findings to real-world scenarios while mitigating potential biases and improving the robustness of our methodology.”
Round 2
Reviewer 1 Report
Comments and Suggestions for Authors
Thanks you for the improvements in your article.
Regarding content, you substantially enriched the related work and the discussion sections which makes the paper stronger.
Personally, I fin the paper is still on the longish side, but if the editor in chief has no concerns, I am following.
However, there are a few typos and grammar issues in the paper that might have been introduced in the rush of making the revisions.
Please check, for instance, the following:
p. 4, line 128: -> emphazises
p. 5, line 151 -> conducted an experiment …
p. 6, line 192 & 198: the verb is missing
p. 6, line 200: spell out QS
p. 6, line 205 f.: this sentence needs rephrasing
p. 22, line 672: this sentence needs rephrasing
Please check the document carefully for capitalisation, spaces, and punctuations. There are still several flaws-
From my side, with a thorough language check, the paper is ready for publication.
Comments on the Quality of English Language
See my comments above. In general, the paper still exhibits a few typos and grammar issues that might have been introduced in the rush of making the revisions. These should be solved in a thorough language check of the paper.
Author Response
Dear Reviewer 1:
Thank you for your thorough review and constructive feedback on the revised manuscript. We have carefully reviewed the specific points you raised and made the corrections to improve the clarity and accuracy of the paper. Additionally, we used Grammarly Business for a comprehensive language check, addressing grammatical, capitalization, spacing, and punctuation issues to enhance the overall quality of the manuscript.
While using LaTeX, we faced an issue with word highlighting causing some lines to extend beyond the margin. But if the highlight has been removed in LaTeX, the lines that previously extended beyond the margin now align appropriately, ensuring that the manuscript adheres to formatting guidelines.
Thank you once again for your valuable feedback and support.